# Almost Surely Asymptotically Constant Graph Neural Networks

**Sam Adam-Day**[*]     **Michael Benedikt**     **İsmail İlkan Ceylan**     **Ben Finkelshtein**
Department of Computer Science
University of Oxford
Oxford, UK

## Abstract

We present a new angle on the expressive power of graph neural networks (GNNs) by studying how the predictions of real-valued GNN classifiers, such as those classifying graphs probabilistically, evolve as we apply them on larger graphs drawn from some random graph model. We show that the output converges to a *constant* function, which upper-bounds what these classifiers can uniformly express. This strong convergence phenomenon applies to a very wide class of GNNs, including state of the art models, with aggregates including mean and the attention-based mechanism of graph transformers. Our results apply to a broad class of random graph models, including sparse and dense variants of the Erdős–Rényi model, the stochastic block model, and the Barabási-Albert model. We empirically validate these findings, observing that the convergence phenomenon appears not only on random graphs but also on some real-world graphs.

## 1 Introduction

Graph neural networks (GNNs) [44, 21] have become prominent for graph machine learning with applications in domains such as life sciences [47, 11, 25, 55]. Their empirical success motivated work investigating their theoretical properties, pertaining to their expressive power [52, 39, 4, 8, 1, 43, 23, 19, 54], generalization capabilities [18, 33, 38, 45, 40], and convergence properties [28, 27, 38, 32, 2].

We consider GNNs outputting real-valued vectors, such as those which classify graphs probabilistically, and ask: *how do the outputs of these GNNs evolve as we apply them on larger graphs drawn from a random graph model?*

Our study provides a surprising answer to this question: the output of many GNNs eventually

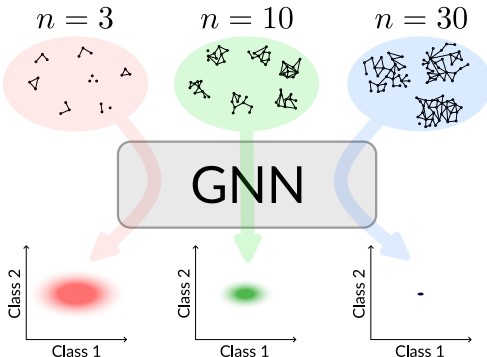

Figure 1: The output of the considered GNNs eventually become constant as the graph sizes increase.

become *independent* of their inputs – each model eventually outputs the same values on all graphs – as graph sizes increase (Figure 1). This "almost sure convergence" to a *constant distribution* is much stronger than convergence to *some* limit object [28, 27, 32]. The immediate consequence of this strong convergence phenomenon is to upper bound the uniform expressiveness of the considered model architectures: *these architectures can uniformly express only the classifiers that are almost*

---

[*]Corresponding author. Email: `me@samadamday.com`

38th Conference on Neural Information Processing Systems (NeurIPS 2024).

*surely asymptotically constant.* In other words, our results provide impossibility results for what tasks are *in principle* learnable by GNNs. While our top-level results are for graph classification, in the process we provide strong limitations on what node- and edge-classification can be performed by GNNs on random graphs: see, for example Theorem 5.3.

**Scope of the result.** The core approach in graph machine learning is based on iteratively updating node representations of an input graph by an *aggregate* of messages flowing from the node's neighbours [20]. This approach can be extended to global aggregates [5]. Our main result holds for all architectures that use *weighted mean* as an aggregation function and it is extremely robust in the following two dimensions:

1. *Model architectures*: Our result is very general and abstracts away from low-level architectural design choices. To achieve this, we introduce an *aggregate term language* using weighted mean aggregation and provide an "almost sure optimization" result for this language: our result states that every term in the language can be simplified to Lipschitz functions for most inputs. Thus, any architecture that can be expressed in this language follows the same convergence law. This includes *graph attention networks* (GATs) [49], as well as popular (graph) transformers, such as the *General, Powerful, Scalable Graph Transformer* (GPS) with random walk encodings [41]. The term language can seamlessly capture common design choices, such as skip and jumping knowledge connections [51], or global aggregation schemes [5].

2. *Random graph models*: All results apply to a wide class of random graph models, including Erdős-Rényi models of various sparsity levels, the Barabási-Albert preferential attachment model, and the stochastic block model. The sparse models are more realistic than their dense counterparts which makes it typically harder to obtain results for them. This is also reflected in our study, as the results for sparse and dense models require very different proofs.

**Contributions.** The key contributions of this paper are as follows:

- We introduce a flexible aggregate term language with attractive closure properties (Section 4) and prove an "almost sure convergence" result for this language relative to a wide class of random graph models (Section 5). This result is of independent interest since it pushes the envelope for convergence results from classical logical languages [15, 36] to include aggregation.

- We show that a diverse class of architectures acting as real-valued graph classifiers can be expressed in the term language (Section 4). In Section 5 we present "almost sure convergence" results for our term language, from which we derive results about GNNs (Corollary 5.2). The results are robust to many practical architectural design choices and even hold for architectures using mixtures of layers from different architectures. We also show strong convergence results for real-valued node classifiers in many graph models.

- We validate these results empirically, showing the convergence of these graph classifiers in practice (Section 6) on graphs drawn from the random models studied. In addition, we probe the real-world significance of our results by testing for convergence on a dataset with varying size dataset splits. Across all experiments we observe rapid convergence to a constant distribution. Interestingly, we note some distinctions between the convergence of the sparse and non-sparse Erdős-Rényi model, which we can relate to the proof strategies for our convergence laws.

## 2 Related work

**Uniform expressiveness.** The expressive power of MPNNs is studied from different angles, including their power in terms of graph distinguishability [52, 39]. The seminal results of Xu et al. [52], Morris et al. [39] show that MPNNs are upper bounded by the *1-dimensional Weisfeiler Leman graph isomorphism test (1-WL)* in terms of graph distinguishability. WL-style expressiveness results are inherently non-uniform, i.e., the model construction is dependent on the graph size. There are also recent studies that focus on uniform expressiveness [4, 42, 2]. In particular, Adam-Day et al. [2] investigate the uniform expressive power of GNNs with randomized node features, which are known to be more expressive in the non-uniform setting [1, 43]. They show that for classical Erdős-Rényi graphs, GNN binary classifiers display a zero-one law, assuming certain restrictions on GNN weights and the random graph model. We focus on real-valued classifiers, where their results do not apply, while dealing with a wider class of random graph models, subsuming popular architectures such as graph transformers.

**Convergence laws for languages.** Our work situates GNNs within a rich term language built up from graph and node primitives via real-valued functions and aggregates. Thus it relates to convergence laws for logic-based languages on random structures, dating back to the zero-one law of Fagin [15], including [35, 46, 31]. We are not aware of *any* prior convergence laws for languages with aggregates; the only work on numerical term languages is by Grädel et al. [22], which deals with a variant of first-order logic in general semi-rings.

**Other notions of convergence on random graphs.** The works of Cordonnier et al. [9], Keriven et al. [28, 27], Maskey et al. [38], Levie [32] consider convergence to continuous analogues of GNNs, often working within metrics on a function space. The results often focus on dense random graph models, such as graphons [34]. Our approach is fundamentally different in that we can use the standard notion of asymptotic convergence in Euclidean space, comparable to traditional language-based convergence results outlined above, such as those by Fagin [15] and Lynch [36]. The key point is that a.a.s. constancy is a *very* strong notion of convergence and it does not follow from convergence in the senses above. In fact, obtaining such a strong convergence result depends heavily on the details of the term language, as well as the parameters that control the random graph: see Section 7 for further discussion of the line between a.a.s. convergence and divergence. Our study gives particular emphasis to sparse random graph models, like Barabási-Albert, which are closer to graphs arising in practice.

## 3 Preliminaries

### 3.1 Featured random graphs and convergence

**Random graphs.** We consider simple, undirected graphs $G = (V_G, E_G, H_G)$ where each node is associated with a vector of *node features* given by $H_G : V_G \to \mathbb{R}^d$. We refer to this as a *featured graph*. We are interested in *random graph models*, specifying for each number $n$ a distribution $\mu_n$ on graphs with $n$ nodes, along with *random graph feature models*, where we have a distribution $\mu_n$ on featured graphs with $n$ nodes. Given a random graph model and a distribution $\nu$ over $\mathbb{R}^d$, we get a random graph feature model by letting the node features be chosen independently of the graph structure via $\nu$.

**Erdős-Rényi and the stochastic block model.** The most basic random graph model we deal with is the *Erdős-Rényi distribution* $\mathrm{ER}(n, p(n))$, where an edge is included in the graph with $n$ nodes with probability $p(n)$ [14]. The classical case is when $p(n)$ is independent of the graph size $n$, which we refer as the *dense* ER distribution. We also consider the *stochastic block model* $\mathrm{SBM}(n_1, \ldots, n_M, P)$, which contains $m$ communities of sizes $n_1, \ldots, n_M$ and an edge probability matrix between communities $\boldsymbol{P} \in \mathbb{R}^{M \times M}$. A community $i$ is sampled from the Erdős-Rényi distribution $\mathrm{ER}(n, p(n) = \boldsymbol{P}_{i,i})$ and an edge to a node in another community $j$ is included with probability $\boldsymbol{P}_{i,j}$.

**The Barabási-Albert preferential attachment model.** Many graphs encountered in the real world obey a *power law*, in which a few vertices are far more connected than the rest [7]. The *Barabási-Albert distribution* was developed to model this phenomenon [3]. It is parametrised by a single integer $m$, and the $n$-vertex graph $\mathrm{BA}(n, m)$ is generated sequentially, beginning with a fully connected $m$-vertex graph. Nodes are added one at a time and get connected via $m$ new edges to previous nodes, where the probability of attaching to a node is proportional to its degree.

**Almost sure convergence.** Given any function $F$ from featured graphs to real vectors and a random featured graph model $(\mu_n)_{n \in \mathbb{N}}$, we say $F$ *converges asymptotically almost surely* (converges a.a.s.) to a vector $\boldsymbol{z}$ with respect to $\bar{\mu}$ if for all $\epsilon, \theta > 0$ there is $N \in \mathbb{N}$ such that for all $n \geq N$, with probability at least $1 - \theta$ when drawing featured graphs $G$ from $\mu_n$, we have that $\|F(G) - \boldsymbol{z}\| < \epsilon$.

### 3.2 Graph neural networks and graph transformers

We first briefly introduce *message passing neural networks* (MPNNs) [20], which include the vast majority of graph neural networks, as well as (graph) transformers.

**Message passing neural networks.** Given a featured graph $G = (V_G, E_G, H_G)$, an MPNN sets the initial features $\boldsymbol{h}_v^{(0)} = H_G(v)$ and iteratively updates the feature $\boldsymbol{h}_v^{(\ell)}$ of each node $v$, for

$0 \leq \ell \leq L - 1$, based on the node's state and the state of its neighbors $\mathcal{N}(v)$ by defining $\boldsymbol{h}_v^{(\ell+1)}$ as:

$$\text{UPD}^{(\ell)} \left( \boldsymbol{h}_v^{(\ell)}, \text{AGG}^{(\ell)} \left( \boldsymbol{h}_v^{(\ell)}, \{\!\{ \boldsymbol{h}_u^{(\ell)} \mid u \in \mathcal{N}(v) \}\!\} \right) \right),$$

where $\{\!\{ \cdot \}\!\}$ denotes a multiset and $\text{UPD}^{(\ell)}$ and $\text{AGG}^{(\ell)}$ are differentiable *update* and *aggregation* functions, respectively. The final node embeddings are pooled to form a graph embedding vector $\boldsymbol{z}_G^{(L)}$ to predict properties of entire graphs. A MEANGNN is an MPNN where the aggregate is mean.

**GATs.** One class of MPNNs are graph attention networks (GATs) [49], where each node is updated with a weighted average of its neighbours' representations, letting $\boldsymbol{h}_v^{(\ell+1)}$ be:

$$\sum_{u \in \mathcal{N}(v)} \frac{\exp\left( \text{score}\left( \boldsymbol{h}_v^{(\ell)}, \boldsymbol{h}_u^{(\ell)} \right) \right)}{\sum_{w \in \mathcal{N}(v)} \exp\left( \text{score}\left( \boldsymbol{h}_v^{(\ell)}, \boldsymbol{h}_w^{(\ell)} \right) \right)} \boldsymbol{W}^{(\ell)} \boldsymbol{h}_u^{(\ell)},$$

where $\text{score}$ is a certain learnable Lipschitz function.

**Graph transformers.** Beyond traditional MPNNs, *graph transformers* extend the well-known transformer architecture to the graph domain. The key ingredient in transformers is the self-attention mechanism. Given a featured graph, a single attention head computes a new representation for every (query) node $v$, in every layer $\ell > 0$ as follows:

$$\text{ATT}(v) = \sum_{u \in V_G} \frac{\exp\left( \text{scale}\left( \boldsymbol{h}_v^{(\ell)}, \boldsymbol{h}_u^{(\ell)} \right) \right)}{\sum_{w \in V_G} \exp\left( \text{scale}\left( \boldsymbol{h}_v^{(\ell)}, \boldsymbol{h}_w^{(\ell)} \right) \right)} \boldsymbol{W}^{(\ell)} \boldsymbol{h}_u^{(\ell)}$$

where $\text{scale}$ is another learnable Lipschitz function (the scaled dot-product).

The vanilla transformer architecture ignores the graph structure. Graph transformer architectures [53, 41, 37] address this by explicitly encoding graph inductive biases, most typically in the form of positional encodings (PEs). In their simplest form, these encodings are additional features $\boldsymbol{p}_v$ for every node $v$ that encode a node property (e.g., node degree) which are concatenated to the node features $\boldsymbol{h}_v$. The random walk positional encoding (RW) [12] of each node $v$ is given by:

$$\text{rw}_v = [\text{rw}_{v,1}, \text{rw}_{v,2}, \dots \text{rw}_{v,k}],$$

where $\text{rw}_{v,i}$ is the probability of an $i$-length random walk that starts at $v$ to end at $v$.

The GPS architecture [41] is a representative graph transformer, which applies a parallel computation in every layer: a transformer layer (with or without PEs) and an MPNN layer are applied in parallel and their outputs are summed to yield the node representations. By including a standard MPNN in this way, a GPS layer can take advantage of the graph topology even when there is no positional encoding. In the context of this paper, we write GPS to refer to a GPS architecture that uses an MPNN with *mean* aggregation, and GPS+RW if the architecture additionally uses a random-walk PE.

**Probabilistic classifiers.** We are looking at models that produce a vector of reals on each graph. All of these models can be used as probabilistic graph classifiers. We only need to ensure that the final layer is a softmax or sigmoid applied to pooled representations.

## 4 Model architectures via term languages

We demonstrate the robustness and generality of the convergence phenomenon by defining a term language consisting of compositions of operators on graphs. Terms are formal sequences of symbols which when interpreted in a given graph yield a real-valued function on the graph nodes.

**Definition 4.1** (Term language). $\text{AGG}[\text{WMEAN}]$ is a term language which contains node variables $x, y, z, \dots$ and terms defined inductively:[2]

- The *basic terms* are of the form $\text{H}(x)$, representing the features of the node $x$, and constants $\boldsymbol{c}$.

---

[2]In the body we reserve $x, y, z$ for free variables and $u, v, w$ for concrete nodes in a graph.

- Let $h\colon \mathbb{R}^d \to (0,\infty)^d$ be a function which is Lipschitz continuous on every compact domain. Given terms $\tau$ and $\pi$, the *local h-weighted mean* for node variable $x$ is:

$$\sum_{y \in \mathcal{N}(x)} \tau(y) \star h(\pi(y))$$

  The interpretation of $\star$ will be defined below. The *global h-weighted mean* is the term:

$$\sum_{y} \tau(y) \star h(\pi(y))$$

- Terms are closed under applying a function symbol for each Lipschitz continuous $F\colon \mathbb{R}^{d \times k} \to \mathbb{R}^d$ for any $k \in \mathbb{N}^+$.

The weighted mean operator takes a weighted average of the values returned by $\tau$. It uses $\pi$ to perform a weighting, normalizing the values of $\pi$ using $h$ to ensure that we are not dividing by zero (see below for the precise definition).

To avoid notational clutter, we keep the dimension of each term fixed at $d$. It is possible to simulate terms with different dimensions by letting $d$ be the maximum dimension and padding the vectors with zeros, noting that the padding operation is Lipschitz continuous.

We make the interpretation of the terms precise as follows. See Figure 2 for a graphical example of evaluating a term on a graph.

**Definition 4.2.** Let $G = (V_G, E_G, H_G)$ be a featured graph. Let $\tau$ be a term with free variables $x_1, \ldots, x_k$ and $\bar{u} = (u_1, \ldots, u_k)$ a tuple of nodes. The *interpretation* $[\![\tau(\bar{u})]\!]_G$ of term $\tau$ graph $G$ for tuple $\bar{u}$ is defined recursively:

- $[\![c(\bar{u})]\!]_G = c$ for any constant $c$.

- $[\![\mathrm{H}(x_i)(\bar{u})]\!]_G = H_G(u_i)$ for the $i^{\text{th}}$ node's features.

- $[\![F(\tau_1, \ldots, \tau_k)(\bar{u})]\!]_G = F([\![\tau_1(\bar{u})]\!]_G, \ldots, [\![\tau_k(\bar{u})]\!]_G)$ for any function symbol $F$.

- For any term composed using $\star$:

$$\left[\!\!\left[ \sum_{y \in \mathcal{N}(x_i)} \tau \star h(\pi)(\bar{u}) \right]\!\!\right]_G = \begin{cases} \dfrac{\sum\limits_{v \in \mathcal{N}(u_i)} [\![\tau(\bar{u},v)]\!]_G \, h([\![\pi(\bar{u},v)]\!]_G)}{\sum\limits_{v \in \mathcal{N}(u_i)} h([\![\pi(\bar{u},v)]\!]_G)} & \text{if } \mathcal{N}(u_i) \neq \emptyset; \\[4ex] 0 & \text{if } \mathcal{N}(u_i) = \emptyset. \end{cases}$$

The semantics of global weighted mean is defined analogously and omitted for brevity.

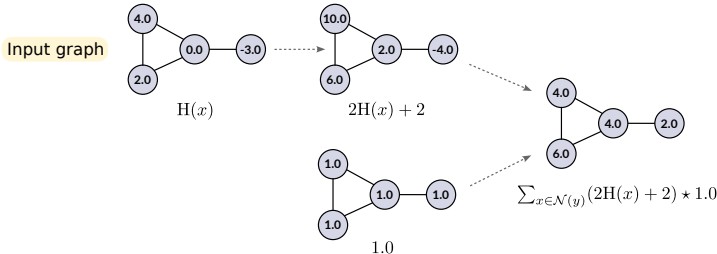

Figure 2: Evaluation of the term $\sum_{x \in \mathcal{N}(y)} (2\mathrm{H}(x) + 2) \star 1.0$ on a small graph with scalar features. The term computes the mean of $z \mapsto 2z + 2$ on each of a node's neighbours. As each sub-term has one free variable, we can represent the intermediate results as scalar values for each node.

A *closed* term has all node variables bound by a weighted mean operator: so the implicit input is just a featured graph.

**Definition 4.3.** We augment the term language to $\mathrm{AGG}[\textsc{Wmean}, \textsc{RW}]$ by adding the random walk operator $\mathrm{rw}(x)$. The interpretation of $\mathrm{rw}(x_i)$ given a graph $G$ and a tuple of nodes $\bar{u}$, is:

$$[\![\mathrm{rw}(x_i)(\bar{u})]\!]_G = [\mathrm{rw}_{u_i,1}, \ldots, \mathrm{rw}_{u_i,d}]$$

### 4.1 How powerful is the term language?

Various architectures can be described using this term language. The core idea is always the same: we show that all basic building blocks of the architecture can be captured in the term language and applying this inductively yields the desired result. Let us first note that all linear functions and all commonly used activation functions are Lipschitz continuous, and therefore included in the language.

**MPNNs with mean aggregation.** Consider an $L$-layer MPNN with mean aggregation, update functions $\textsc{Upd}^{(\ell)}$ consisting of an activation function applied to a linear transformation, with mean pooling at the end. First, note that mean aggregation can be expressed as:

$$\textsc{Mean}_y \pi(y) := \sum_y \pi(y) \star 1.$$

For each layer $0 \leq \ell < L$, we define a term $\tau^{(\ell)}(x)$ which will compute the representation of a node at layer $\ell$ of the MPNN:

- **Initialization**. Let $\tau^{(0)}(x) := \mathrm{H}(x)$. Then the value $[\![\tau^{(0)}(x)(u)]\!]_G$ at a node $u$ is the initial node representation $H(u)$.
- **Layers**. For $1 \leq \ell < L$, define $\tau^{(\ell+1)}(x) := \textsc{Upd}^{(\ell)}\left(x, \textsc{Mean}_{y \in \mathcal{N}(x)} \tau^{(\ell)}(y)\right)$. Then the value at node $u$ is the following, which conforms with the inductive construction of the MPNN:

$$[\![\tau^{(\ell+1)}(x)(u)]\!]_G = \textsc{Upd}^{(\ell)}\left([\![\tau^{(\ell)}(x)(u)]\!]_G, \frac{1}{|\mathcal{N}(u)|} \sum_{v \in \mathcal{N}(u)} [\![\tau^{(\ell)}(x)(v)]\!]_G\right)$$

- **Final mean pooling**. The final graph representation is computed as $\tau := \textsc{Mean}_x \tau^{(L)}(x)$.

The idea is similar for the other architectures, where the difference lies in the aggregation functions. Thus below we only present how the term language captures their respective aggregation functions.

**Graph transformers.** We can express the self-attention mechanism of transformers using the following aggregator:

$$\textsc{Att}_y \pi(y) := \sum_y \pi(y) \star \exp(\mathrm{scale}(\pi(x), \pi(y)))$$

The function $\mathrm{scale}(\pi(x), \pi(y))$ is a term in the language, since scaled dot product attention is a Lipschitz function. To see how graph transformer architectures such as GPS can be expressed, it suffices to note that we can express both self-attention layers and MPNN layers with mean aggregation, since the term language is closed under addition. The random walk positional encoding can also be expressed using the rw operator.

**Graph attention networks.** The attention mechanism of GAT is local to a node's neighbours and can be expressed in our term language using similar ideas, except using the local aggregate terms.

**Additional architectural features.** Because the term language allows the arbitrary combination of graph operations, it can robustly capture many common architectural choices used in graph learning architectures. For example, a *skip connection* or *residual connection* from layer $\ell_1$ to layer $\ell_2$ can be expressed by including a copy of the term for layer $\ell_1$ in the term for the layer $\ell_2$ [51]. *Global readout* can be captured using a global mean aggregation [5]. Attention *conditioned on computed node or node-pair representations* can be captured by including the term which computes these representations in the mean weight [37].

**Capturing probabilistic classification.** Our term language defines bounded vector-valued functions over graphs. Standard normalization functions, like softmax and sigmoid, are easily expressible in our term language, so probabilistic classifiers are subsumed.

**Graph convolutional networks.** In Appendix A we show how to extend the term language to incorporate graph convolutional networks (GCNs) [29] by adding a new aggregator.

## 5 Convergence theorems

We start by presenting the convergence theorem.

**Theorem 5.1.** *Consider $(\mu_n)_{n \in \mathbb{N}}$ sampling a graph $G$ from any of the following models and node features independently from i.i.d. bounded distributions on $d$ features.*

1. *The Erdős-Rényi distribution $\mathrm{ER}(n, p(n))$ where $p$ satisfies any of the following properties.*
   - ***Density.*** *$p$ converges to $\tilde{p} > 0$.*
   - ***Root growth.*** *For some $K > 0$ and $0 < \beta < 1$ we have: $p(n) = Kn^{-\beta}$.*
   - ***Logarithmic growth.*** *For some $K > 0$ we have: $p(n) = K \frac{\log(n)}{n}$.*
   - ***Sparsity.*** *For some $K > 0$ we have: $p(n) = Kn^{-1}$.*

2. *The Barabási-Albert model $\mathrm{BA}(n, m)$ for any $m \geq 1$.*

3. *The stochastic block model $\mathrm{SBM}(n_1(n), \ldots, n_m(n), \boldsymbol{P})$ where $n_1, \ldots, n_m \colon \mathbb{N} \to \mathbb{N}$ are such that $n_1(n) + \cdots + n_m(n) = n$ and each $\frac{n_i}{n}$ converges, and $\boldsymbol{P}$ is any symmetric $m \times m$ edge probability matrix.*

*Then every $\mathrm{AGG}[\mathrm{WMEAN}, \mathrm{RW}]$ term converges a.a.s. to a constant with respect to $(\mu_n)_{n \in \mathbb{N}}$.*[3]

Concretely, this result shows that for any probabilistic classifier, or other real-valued classifier, which can be expressed within the term language, when drawing graphs from any of these distributions, eventually the output of the classifier will be the same regardless of the input graph, asymptotically almost surely. Thus, the only probabilistic classifiers which can be expressed by such models are those which are asymptotically constant.

**Corollary 5.2.** *For any of the random graph featured models above, for any MeanGNN, GAT, or GPS + RW, there is a distribution $\bar{p}$ on the classes such that the class probabilities converge asymptotically almost surely to $\bar{p}$.*

We now discuss briefly how the results are proven. The cases divide into two groups: the denser cases (the first three ER distributions and the SBM) and the sparser cases (the fourth ER distribution and the BA model). Each is proved with a different strategy.

## 5.1 Overview of the technical constructions for the denser cases

While the theorem is about closed terms, naturally we need to prove it inductively on the term language, which requires consideration of terms with free variables. We show that each open term in some sense degenerates to a Lipschitz function almost surely. The only caveat is that we may need to distinguish based on the "type" of the node – for example, nodes $u_1, u_2, u_3$ that form a triangle may require a different function from nodes that do not. Formally, for node variables $\bar{x}$, an $\bar{x}$ *graph type* is a conjunction of expressions $E(x_i, x_j)$ and their negations. The *graph type of tuple $\bar{u}$ in a graph*, denoted $\mathrm{GrTp}(\bar{u})$ is the set of all edge relations and their negations that hold between elements of $\bar{u}$. A $(\bar{x}, d)$ *feature-type controller* is a Lipschitz function taking as input pairs consisting of $d$-dimensional real vector and an $\bar{x}$ graph type.

The key theorem below shows that to each term $\pi$ we can associate a feature-type controller $e_\pi$ which captures the asymptotic behaviour of $\pi$, in the sense that with high probability, for most of the tuples $\bar{u}$ the value of $e_\pi(H_G(\bar{u}), \mathrm{GrTp}(\bar{u}))$ is close to $[\![\pi(\bar{u})]\!]$.

**Theorem 5.3** (Aggregate Elimination for Non-Sparse Graphs). *For all terms $\pi(\bar{x})$ over featured graphs with $d$ features, there is a $(\bar{x}, d)$ feature-type controller $e_\pi$ such that for every $\epsilon, \delta, \theta > 0$, there is $N \in \mathbb{N}$ such that for all $n \geq N$, with probability at least $1 - \theta$ in the space of graphs of size $n$, out of all the tuples $\bar{u}$ at least $1 - \delta$ satisfy that $\|e_\pi(H_G(\bar{u}), \mathrm{GrTp}(\bar{u})) - [\![\pi(\bar{u})]\!]\| < \epsilon$.*

This can be seen as a kind of "almost sure quantifier elimination" (thinking of aggregates as quantifiers), in the spirit of Kaila [24], Keisler and Lotfallah [26]. It is proven by induction on term depth, with the log neighbourhood bound playing a critical role in the induction step for weighted mean.

Theorem 5.3 highlights an advantage of working with a term language having nice closure properties, rather than directly with GNNs: it allows us to use induction on term construction, which may be more natural and more powerful than induction on layers. Theorem 5.3 also gives strong limitations on *node and link classification* using GNNs: on most nodes in (non-sparse) random graphs, GNNs can only classify based on the features of a node, they cannot make use of any graph structure.

---

[3]The appendix includes additional results for the GCN aggregator.

## 5.2 Overview of the technical constructions for the sparser cases

In the sparser cases, the analysis is a bit more involved. Instead of graph types over $\bar{u}$, which only specify graph relations among the $\bar{u}$, we require descriptions of local neighbourhoods of $\bar{u}$.

**Definition 5.4.** Let $G$ be a graph, $\bar{u}$ a tuple of nodes in $G$ and $\ell \in \mathbb{N}$. The *$\ell$-neighbourhood of $\bar{u}$ in $G$*, denoted $\mathcal{N}_\ell(\bar{u})$ is the subgraph of $G$ induced by the nodes of distance at most $\ell$ from some node in $\bar{u}$.

The "types" are now graphs $T$ with $k$ distinguished elements $\bar{w}$, which we call *$k$-rooted graphs*. Two $k$-rooted graphs $(T, \bar{w})$ and $(U, \bar{z})$ are *isomorphic* is there is a structure-preserving bijection $T \to U$ which maps $\bar{w}$ to $\bar{z}$. The combinatorial tool here is a fact known in the literature as 'weak local convergence': the percentage of local neighbourhoods of any given type converges.

**Lemma 5.5** (Weak local convergence). *Consider sampling a graph $G$ from either the sparse* ER *or* BA *distributions. Let $(T, \bar{w})$ be a $k$-rooted graph and take $\ell \in \mathbb{N}$. There is $q_T \in [0, 1]$ such that for all $\epsilon, \theta > 0$ there is $N \in \mathbb{N}$ such that for all $n \geq N$ with probability at least $1 - \theta$ we have that:*

$$\left| \frac{|\{\bar{u} \mid \mathcal{N}_\ell(\bar{u}) \cong (T, \bar{w})\}|}{n} - q_T \right| < \epsilon$$

Our "aggregate elimination", analogous to Theorem 5.3, states that every term can be approximated by a Lipschitz function of the features and the neighbourhood type.

**Theorem 5.6** (Aggregate Elimination for Sparser Graphs). *For every $\pi(\bar{x})$, letting $\ell$ be the maximum aggregator nesting depth in $\pi$, for all $k$-rooted graphs $(T, \bar{w})$ there is a Lipschitz function $e_\pi^T : \mathbb{R}^{|T| \cdot d} \to \mathbb{R}^d$ such that for each $\epsilon, \theta > 0$, there is $N \in \mathbb{N}$ such that for all $n \geq N$ with probability at least $1 - \theta$ in the space of graphs of size $n$, for every $k$-tuple of nodes $\bar{u}$ in the graph such that $\mathcal{N}_\ell(\bar{u}) \cong (T, \bar{w})$ we have that $\left\| e_\pi^T(H_G(\mathcal{N}_\ell(\bar{u}))) - [\![\pi(\bar{u})]\!] \right\| < \epsilon$.*

Compared to Theorem 5.3 the result is much less limiting in what node-classifying GNNs can express. Although combining the sparse and non-sparse conditions covers many possible growth rates, it is not true that one gets convergence for Erdős-Rényi with *arbitrary* growth functions:

**Theorem 5.7.** *There are functions $p(n)$ converging to zero and a term $\tau$ in our language such that $\tau$ does not converge even in distribution (and hence does not converge a.a.s.) over ER random graphs with growth rate $p$.*

*Proof.* Let $p$ alternate between $\frac{1}{2}$ on even $n$ and $\frac{1}{n}$ on odd $n$. Consider $\tau_1(x)$ that returns $0$ if $x$ has a neighbour and $1$ otherwise, and let $\tau$ be the global average of $\tau_1$. So $\tau$ is the percentage of isolated nodes in the graph. Then $\tau$ clearly goes to zero in the non-sparse case. However the probability that a particular node is not isolated in a random sparse graph of size $n$ is $(1 - \frac{1}{n})^{n-1}$, which goes to $\frac{1}{e} < 1$. Thus $[\![\tau]\!]$ diverges on $\mathrm{ER}(n, p(n))$. $\square$

## 6 Experimental evaluation

We first empirically verify our findings on random graphs and then on a real-world graph to answer the following questions: **Q1**. Do we empirically observe convergence? **Q2**. What is the impact of the different weighted mean aggregations on the convergence? **Q3**. What is the impact of the graph distribution on the convergence? **Q4**. Can these phenomena arise within large real-world graphs?

All our experiments were run on a single NVidia GTX V100 GPU. We made our codebase available online at `https://github.com/benfinkelshtein/GNN-Asymptotically-Constant`.

**Setup.** We report experiments for the architectures MeanGNN, GAT [49], and GPS+RW [41] with random walks of length up to 5. Our setup is carefully designed to eliminate confounding factors:

- We consider five models with the same architecture, each having randomly initialized weights, utilizing a ReLU non-linearity, and applying a softmax function to their outputs. Each model uses a hidden dimension of $128$, $3$ layers and an output dimension of $5$.

- We experiment with distributions $\mathrm{ER}(n, p(n) = 0.1)$, $\mathrm{ER}(n, p(n) = \frac{\log n}{n})$, $\mathrm{ER}(n, p(n) = \frac{1}{50n})$, $\mathrm{BA}(n, m = 5)$. We also experiment with an SBM of 10 communities with equal size, where an edge between nodes within the same community is included with probability $0.7$ and an edge between nodes of different communities is included with probability $0.1$.

- We draw graphs of sizes up to 10,000, where we take 100 samples of each graph size. Node features are independently drawn from $U[0, 1]$ and the initial feature dimension is 128.

To understand the behaviour of the respective models, we will draw larger and larger graphs from the graph distributions. We use five different models to ensure this is not a model-specific behaviour. Further experimental results are reported in Appendix F.

## 6.1 Empirical results on random graph models

In Figure 3, a single model initialization of the MeanGNN, GAT and GPS+RW architectures is used with $\mathrm{ER}(n, p(n) = 0.1)$, $\mathrm{ER}(n, p(n) = \frac{\log n}{n})$ and $\mathrm{ER}(n, p(n) = \frac{1}{50n})$. Each curve in the plots corresponds to a different class probability, depicting the average of 100 samples for each graph size along with the standard deviation shown in lower opacity.

The convergence of class probabilities is apparent across all models and graph distributions, as illustrated in Figure 3, in accordance with our main theorems (**Q1**). The key differences between the plots are the convergence time, the standard deviation and the converged values.

One striking feature of Figure 3 is that the eventual constant output of each model is the same for the dense and logarithmic growth distributions, but not for the sparse distribution (**Q3**). We can relate this to the distinct proof strategy employed in the sparse case, which uses convergence of the proportions of local isomorphism types. There are many local isomorphism types, and the experiments show that what we converge to depends the proportion of these. In all other cases the neighbourhood sizes are unbounded, so there is asymptotically almost surely one 'local graph type'.

We observe that attention-based models such as GAT and GPS+RW exhibit delayed convergence and greater standard deviation in comparison to MeanGNN (**Q2**). A possible explanation is that because some nodes are weighted more than others, the attention aggregation has a higher variance than regular mean aggregation. For instance, if half of the nodes have weights close to 0, then the

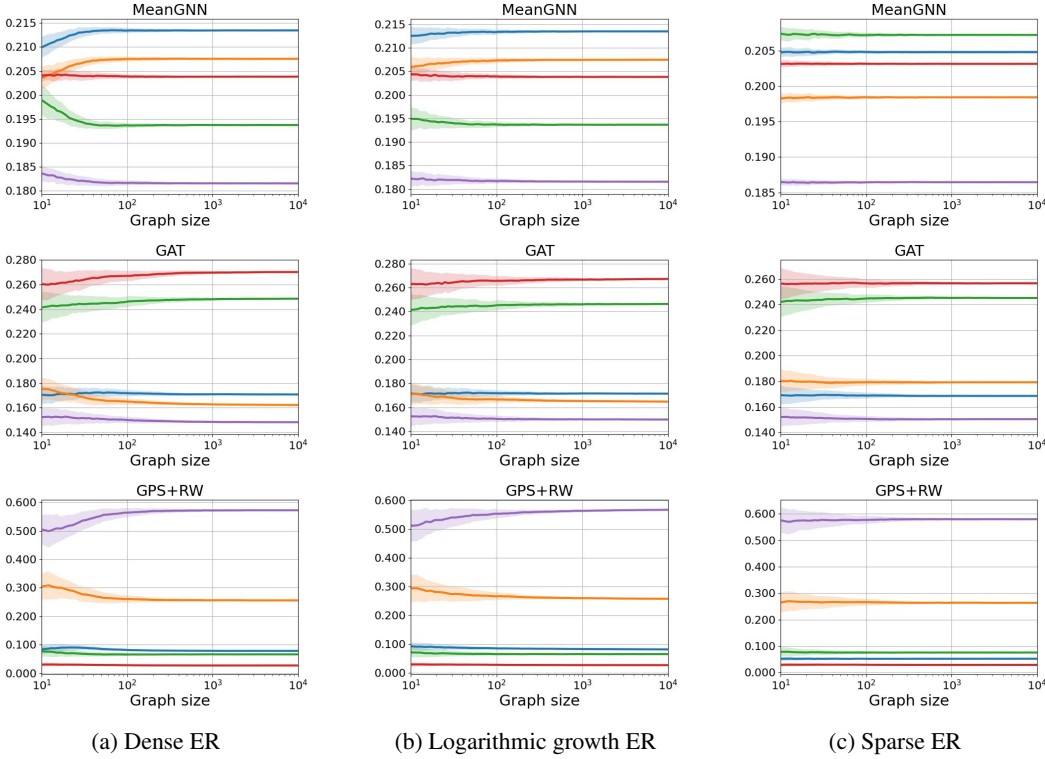

(a) Dense ER       (b) Logarithmic growth ER       (c) Sparse ER

Figure 3: Each plot shows the five mean class probabilities (in different colours) with standard deviations of a single model initialization over $\mathrm{ER}(n, p(n) = 0.1)$, $\mathrm{ER}(n, p(n) = \frac{\log n}{n})$, and $\mathrm{ER}(n, p(n) = \frac{1}{50n})$, as we draw graphs of increasing size.

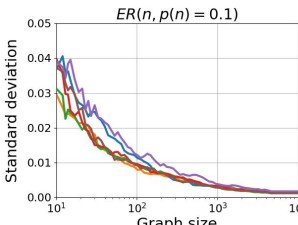
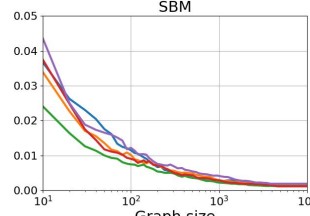
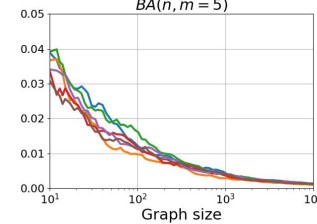

Figure 4: Each plot depicts the standard deviation of Euclidean distances between class probabilities and their respective means across various samples of each graph size for GPS+RW.

attention aggregation effectively takes a mean over half of the available nodes. Eventually, however, the attention weights themselves converge, and thus convergence cannot be postponed indefinitely.

Figure 4 depicts the outcomes of the GPS+RW architecture for various graph distributions. The analysis involves calculating the Euclidean distance between the class probabilities of the GPS+RW architecture and the mean class probabilities over the different graph samples. The standard deviation across the different graph samples is then derived for each of the 5 different model initializations and presented in Figure 4. The decrease of standard deviation across the different model initializations in Figure 4 indicates that all class probabilities converge across the different model initializations, empirically verifying the phenomenon for varying model initializations (**Q1** and **Q3**).

## 6.2 Empirical results on large real-world graphs

Towards **Q4**, we investigated a large real-world dataset. Many commonly studied graph datasets (e.g. ZINC [13], QM9 [6]) do not exhibit sufficient graph size variance and provide no obvious means to add scaling. We used the *TIGER-Alaska* dataset [16] of geographic faces. The original dataset has 93366 nodes, while Dimitrov et al. [10] extracted smaller datasets with graphs having 1K, 5K, 10K, 25K and 90K nodes. We chose the modified dataset as it is split by graph size, and consists of graphs differing from our random models (in particular all graphs are planar). Figure 5 shows the results of applying the same five MeanGNNs to graphs of increasing sizes. Strikingly, we again observe a convergence phenomenon, but at a slower pace.

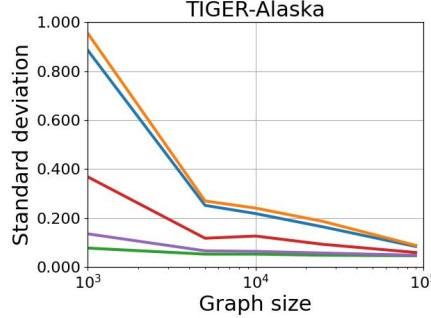

Figure 5: Standard deviation of distances between class probabilities and their means across TIGER-Alaska graph sizes for MeanGNN.

## 7 Discussion and limitations

We have demonstrated a wide convergence phenomenon for real-valued classifiers expressed even in very advanced GNN architectures, and it applies to a great variety of random graph models. Rather than having separate proof techniques per GNN model, our paper introduces a broad language where such models can be situated, and provides techniques at the level of term languages. Although our top-level theorems deal with graph-level tasks, along the way we provide strong limitative results on what can be achieved on random graphs for node- or edge-level real-valued tasks: see Theorem 5.3.

The principal limitations of our work come our assumptions. In particular, we assume that the initial node embeddings are i.i.d. This assumption is used in the application concentration inequalities throughout the proofs, so loosening it would require careful consideration.

Our main results show that many GNN architectures cannot distinguish large graphs. To overcome this limitation, one could consider moving beyond our term language. For example, if we add *sum aggregation*, the term values clearly diverge, and similarly if we allow non-smooth functions, such as linear inequalities. Further we emphasize that a.a.s. convergence is not universal for our term language, and it does not hold even for ER with *arbitrary* $p(n)$ going to $0$: see Theorem 5.7.

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

## A   Graph Convolutional Networks

In the body of the paper, we mentioned that our results can be extended to graph convolutional networks (GCNs) [29]. In this section we show how to extend our term language to capture them. Later in the appendix when providing proof details for our convergence laws, we will utilize the extended language, thus showing the applicability to GCNs.

GCNs are instances of the message passing neural network in which the update function is defined as follows.

$$h_v^{(\ell+1)} = \sum_{u \in \mathcal{N}(v)} \frac{1}{\sqrt{|\mathcal{N}(v)||\mathcal{N}(u)|}} W^{(\ell)} h_u^{(\ell)}$$

To allow our term language to capture GCNs, we add the new aggregator:

$$\text{GCN}_{y \in \mathcal{N}(x)} \tau(y)$$

The language $\text{AGG}[\text{WMEAN}, \text{GCN}, \text{RW}]$ is the result of closing $\text{AGG}[\text{WMEAN}, \text{RW}]$ under this operator. The semantics are extended as follows.

$$\llbracket \text{GCN}_{y \in \mathcal{N}(x_i)} \tau(\bar{u}) \rrbracket_G = \sum_{v \in \mathcal{N}(u_i)} \frac{1}{\sqrt{|\mathcal{N}(u_i)||\mathcal{N}(v)|}} \llbracket \tau(\bar{u}, v) \rrbracket_G$$

With this operator it is possible to capture GCNs in the same way as MeanGNNs are captured (Section 4). Moreover, the extended term language permits arbitrary combinations of the GCN aggregator with other language features.

## B   Concentration Inequalities

Throughout the appendix, we make use of a few basic inequalities.

**Theorem B.1** (Markov's Inequality). *Let $X$ be a positive random variable with finite mean. Then for any $\lambda > 0$ we have:*

$$\mathbb{P}(X \geq \lambda) \leq \frac{\mathbb{E}[X]}{\lambda}$$

*Proof.* See Proposition 1.2.4, p. 8 of [50]. $\qquad\square$

**Theorem B.2** (Chebyshev's Inequality). *Let $X$ be a random variable with finite mean $\mu$ and finite variance $\sigma^2$. Then for any $\lambda > 0$ we have:*

$$\mathbb{P}(|X - \mu| \geq \lambda) \leq \frac{\sigma^2}{\lambda^2}$$

*Proof.* See Corollary 1.2.5, p. 8 of [50]. $\qquad\square$

**Theorem B.3** (Chernoff's Inequality). *Let $X_i$ for $i \leq n$ be i.i.d. Bernoulli random variables with parameter $p$.*

*(1) For any $\lambda > p$ we have:*

$$\mathbb{P}\left(\sum_{i=1}^{n} X_i \geq \lambda\right) \leq e^{-np}\left(\frac{enp}{\lambda}\right)^{\lambda}$$

*(2) For any $\lambda < p$ we have:*

$$\mathbb{P}\left(\sum_{i=1}^{n} X_i \leq \lambda\right) \leq e^{-np}\left(\frac{enp}{\lambda}\right)^{\lambda}$$

*Proof.* See Theorem 2.3.1, p. 17 of [50] and Exercise 2.3.2, p. 18 of [50]. $\qquad\square$

**Theorem B.4** (Hoeffding's Inequality for bounded random variables). *Let $X_i$ for $i \leq n$ be i.i.d. bounded random variables taking values in $[a, b]$ with common mean $\mu$. Then for any $\lambda > 0$ we have:*

$$\mathbb{P}\left(\left|\sum_{i=1}^{n} X_i - n\mu\right| \geq \lambda\right) \leq 2\exp\left(-\frac{2\lambda^2}{n(b-a)^2}\right)$$

*Proof.* See Theorem 2.2.6, p. 16 of [50]. □

**Corollary B.5** (Hoeffding's Inequality for Bernoulli random variables). *Let $X_i$ for $i \leq n$ be i.i.d. Bernoulli random variables with parameter $p$. Then for any $\lambda > 0$ we have:*

$$\mathbb{P}\left(\left|\sum_{i=1}^{n} X_i - np\right| \geq \lambda\right) \leq 2\exp\left(-\frac{2\lambda^2}{n}\right)$$

## C   Proof for Erdős-Rényi distributions, the non-sparse cases

We new begin with the proof of convergence for the first three cases of the Erdős-Rényi distribution in Theorem 5.1, which we restate here for convenience:

**Theorem C.1.** *Consider $(\mu_n)_{n\in\mathbb{N}}$ sampling a graph $G$ from the Erdős-Rényi distribution $\mathrm{ER}(n, p(n))$ and node features independently from i.i.d. bounded distributions on $d$ features, where $p$ satisfies any of the following properties.*

- ***Density**. $p$ converges to $\tilde{p} > 0$.*
- ***Root growth**. For some $K > 0$ and $0 < \beta < 1$ we have: $p(n) = Kn^{-\beta}$.*
- ***Logarithmic growth**. For some $K > 0$ we have: $p(n) = K\frac{\log(n)}{n}$.*

*Then for every $\mathrm{AGG}[\mathrm{WMEAN}, \mathrm{RW}]$ term converges a.a.s. with respect to $(\mu_n)_{n\in\mathbb{N}}$.*

Note that throughout the following we use $u$ and $v$ to denote both nodes and node variables.

### C.1   Combinatorial and growth lemmas

*Remark* C.2. At various points in the following we will make statements concerning particular nodes of an Erdős-Rényi graph without fixing a graph. For example, we state that for any $u$ we have that its degree $d(u)$ is the sum of $n - 1$ Bernoulli random variables. To make sense of such statements we can fix a canonical enumeration of the nodes of every graph and view our node meta-variables as ranging over indices. So we would translate the previous statement as "for any *node index* $u$ the degree of the $u^{\text{th}}$ node in $\mathrm{ER}(n, p(n))$ is the sum of $n - 1$ Bernoulli random variables."

Both the combinatorial and growth lemmas, and the main inductive results themselves require saying that certain events hold with high probability. Inductively, we will need strengthenings of these compared with the statements given in the body. We will need to consider conditional probabilities, where the conditioning is on a conjunction of atomic statements about the nodes.

**Definition C.3.** A $\wedge$-*description* $\phi(\bar{u})$ is a conjunction of statements about the variables $\bar{u}$ of the form $u_i E u_j$, which expresses that there exists an edge between $u_i$ and $u_j$. A $\wedge$-description $u_{i_1} E u_{j_1} \wedge \cdots \wedge u_{i_m} E u_{j_m}$ holds if and only if there is an edge between $u_{i_\ell}$ and $u_{j_\ell}$ for each $\ell$.

The reason for strengthening the results in this way will become clearer when we are proving the local weighted mean induction step $\sum_{v \in \mathcal{N}(u_i)} \rho(\bar{u}, v) \star h(\eta(\bar{u}, v))$. There we will need our auxiliary results to hold conditioned on $v E u_i$, along with any additional requirements coming from previous induction steps.

We will first need a few auxiliary lemmas about the behaviour of random graphs in the different ER models.

We will need to know that in the non-sparse case, neighborhoods are fairly big:

**Lemma C.4** (Log neighbourhood bound). *Let $p \colon \mathbb{N} \to [0, 1]$ satisfy density, root growth or logarithmic growth. There is $R > 0$ such that for all $\delta, \theta > 0$ there is $N \in \mathbb{N}$ such that for all $n \geq N$, with probability at least $1 - \theta$ when drawing graphs from $\mathrm{ER}(n, p(n))$, we have that the proportion of all nodes $u$ such that $|\mathcal{N}(u)| \geq R\log(n)$ is at least $1 - \delta$.*

We prove a stronger result, which allows us finer control over the growth in each of the non-sparse cases. The strengthening will involve the $\wedge$-descriptions of Definition C.3.

**Lemma C.5.** *Take any $\wedge$-description $\phi$ on $k$ variables, and choose $i \leq k$. We consider $k$-tuples of nodes $\bar{u}$ satisfying $\phi$ and the degrees of the $i^{th}$ node $u_i$.*

*(1) Let $p$ satisfy the density condition, converging to $\tilde{p}$. Then for every $\epsilon, \theta > 0$ there is $N \in \mathbb{N}$ such that for all $n \geq N$, with probability at least $1 - \theta$ when drawing graphs from $\mathrm{ER}(n, p(n))$, for all tuples $\bar{u}$ which satisfy $\phi$ we have that:*

$$n(\tilde{p} - \epsilon) < d(u_i) < n(\tilde{p} + \epsilon)$$

*(2) Let $p$ satisfy the root growth condition, with $p = Kn^{-\beta}$. Then for every $R_1 \in (0, 1)$ and $R_2 > 1$ and for every $\theta > 0$ there is $N \in \mathbb{N}$ such that for all $n \geq N$, with probability at least $1 - \theta$ when drawing graphs from $\mathrm{ER}(n, p(n))$, for all tuples $\bar{u}$ which satisfy $\phi$ we have that:*

$$R_1 K n^{1-\beta} < d(u_i) < R_2 K n^{1-\beta}$$

*(3) Let $p$ satisfy the log growth condition, with $p = K\frac{\log(n)}{n}$. Then for every $R_1 \in (0, 1)$ and $R_2 > 1$ and for every $\delta, \theta > 0$ there is $N \in \mathbb{N}$ such that for all $n \geq N$, with probability at least $1 - \theta$ when drawing graphs from $\mathrm{ER}(n, p(n))$, for at least $1 - \delta$ of the tuples $\bar{u}$ which satisfy $\phi$ we have that:*

$$R_1 K \log(n) < d(u_i) < R_2 K \log(n)$$

*Moreover, there is $R_3 > 0$ such that for every $\theta > 0$ there is $N \in \mathbb{N}$ such that for all $n \geq N$, with probability at least $1 - \theta$ when drawing graphs from $\mathrm{ER}(n, p(n))$, for all tuples $\bar{u}$ which satisfy $\phi$ we have that:*

$$d(u_i) < R_3 K \log(n)$$

Note that in the first two cases, we only have claims about all tuples, while in the third case we make an assertion about most tuples, while also asserting an upper bound on all tuples. The upper bound on all tuples does not subsume the upper bound on most tuples, since the former states the existence of an $R_3$, while the latter gives a bound for an arbitrary $R_2$.

We now give the proof of the lemma:

*Proof.*

(1) Take $\epsilon, \theta > 0$. First, there is $N_1$ such that for all $n \geq N_1$:

$$|p(n) - \tilde{p}| < \epsilon$$

Now, any $d(u)$ is the sum of $n - 1$ i.i.d. Bernoulli random variables with parameter $p(n)$: see Remark C.2 for the formalization. Hence by Hoeffding's Inequality (Theorem B.4) and a union bound, there is $N_2$ such that for all $n \geq N_2$, with probability at least $1 - \theta$, for every node $u$ we have:

$$\left|\frac{d(u)}{n} - \tilde{p}\right| < \epsilon$$

Letting $N = \max(N_1, N_2)$, the result follows.

(2) Take $R_1 \in (0, 1)$ and $R_2 > 1$ and fix $\theta > 0$. Again, any $d(u)$ is the sum of $n - 1$ i.i.d. Bernoulli random variables with parameter $p(n)$. By Chernoff's Inequality (Theorem B.3), for any node $u$ we have that:

$$\mathbb{P}\left(d(u) \leq R_1 K n^{1-\beta}\right) \leq \exp\left(K n^{1-\beta}\left(R_1 - 1 - R_1 \log(R_1)\right)\right)$$

Since $R_1 - 1 - R_1 \log(R_1) < 0$, by a union bound there is $N_1$ such that for all $n \geq N_1$, with probability at least $1 - \theta$, for every node $u$ we have:

$$d(u) > R_1 K n^{1-\beta}$$

Similarly there is $N_2$ such that for all $n \geq N_2$, with probability at least $1 - \theta$, for every node $u$ we have:

$$d(u) < R_2 K n^{1-\beta}$$

Letting $N = \max(N_1, N_2)$, the result follows.

(3) Take $R_1 \in (0, 1)$ and $R_2 > 1$ and fix $\delta, \theta > 0$. By Chernoff's Inequality (Theorem B.3), for any node $u$ we have that:

$$\mathbb{P}\left(d(u) \leq R_1 K n^{1-\beta}\right) \leq n^{K(R_1 - 1 - R_1 \log(R_1))}$$

Since $R_1 - 1 - R_1 \log(R_1) < 0$ this probability tends to 0 as $n$ tends to infinity. Hence there is $N_1$ such that for all $n \geq N_1$, for all nodes $u$:

$$\mathbb{P}\left(d(u) \leq R_1 \log(n)\right) < \theta\delta$$

Let $B_n$ be the proportion, our of tuples $\bar{u}$ which satisfy $\phi$, such that:

$$d(u_i) \leq R \log(n)$$

(the 'bad' tuples). Then for $n \geq N_1$, by linearity of expectation $\mathbb{E}[B_n] \leq \theta\delta$.

By Markov's Inequality (Theorem B.1) we have that:

$$\mathbb{P}\left(B_n \geq \delta\right) \leq \frac{\theta\delta}{\delta} = \theta$$

That is, with probability at least $1 - \theta$ we have that the proportionout of tuples $\bar{u}$ which satisfy $\phi$ such that:

$$d(u_i) > R_1 \log(n)$$

is at least $1 - \delta$.

Similarly, there is $N_2$ such that for all $n \geq N_2$, with probability at least $1 - \theta$ we have that the proportion out of tuples $\bar{u}$ which satisfy $\phi$ such that:

$$d(u_i) < R_2 \log(n)$$

is at least $1 - \delta$. Letting $N = \max(N_1, N_2)$ the first statement follows.

To show the 'moreover' part, note that by Chernoff's Inequality (Theorem B.3), for any $R_3 > 0$ and for any node $u$ we have that:

$$\mathbb{P}\left(d(u) \geq R_3 K n^{1-\beta}\right) \leq n^{K(R_3 - 1 - R_3 \log(R_3))}$$

For $R_3$ large enough we have that $K(R_3 - 1 - R_3 \log(R_3)) \leq -2$. Hence taking a union bound for every $\theta > 0$ there is $N \in \mathbb{N}$ such that for all $n \geq N$, with probability at least $1 - \theta$ when drawing graphs from $\text{ER}(n, p(n))$, for all tuples $\bar{u}$ which satisfy $\phi$ we have that:

$$d(u_i) < R_3 K \log(n)$$

$\square$

The basic form of the following "regularity lemma" states that for every $k \in \mathbb{N}^+$ and $i \leq k$, whenever we have a large subset $S$ of the $(k + 1)$-tuples $(\bar{u}, v)$ with each $u_i$ adjacent to $v$, there is a large subset of the $k$-tuples $\bar{u}$ such that each $u_i$ is adjacent to a large set of $v$'s such that $(\bar{u}, v) \in S$. The full regularity lemma states that this is the case also when we restrict the tuples $\bar{u}$ to a certain $\wedge$-description.

**Lemma C.6** (Regularity Lemma). *Let $p \colon \mathbb{N} \to [0, 1]$ satisfy one of the conditions other than sparsity. Take a $\wedge$-description $\phi$ on $k$ variables and fix $i \leq k$. Then for every $\gamma, \theta > 0$ there is $N \in \mathbb{N}$ and $\delta' > 0$ such that for all $n \geq N$ and $\delta < \delta'$, with probability at least $1 - \theta$ we have that, whenever $S \subseteq \{(\bar{u}, v) \mid \phi(\bar{u}) \wedge u_i E v\}$ is such that:*

$$\frac{|S|}{|\{(\bar{u}, v) \mid \phi(\bar{u}) \wedge u_i E v\}|} \geq 1 - \delta$$

*then there is $S' \subseteq \{\bar{u} \mid \phi(\bar{u})$ and $d(u_i) > 0\}$ such that:*

$$\frac{|S'|}{|\{\bar{u} \mid \phi(\bar{u})$ and $d(u_i) > 0\}|} \geq 1 - \gamma$$

*and for every $\bar{u} \in S'$ we have:*

$$\frac{|\{v \mid (\bar{u}, v) \in S\}|}{d(u_i)} \geq 1 - (\delta + \delta^2)$$

For this we make use of the following purely combinatorial lemma.

**Lemma C.7.** *Take $\delta, \sigma > 0$. Let $(B_1, \ldots, B_m)$ be a sequence of disjoint finite sets of minimum size $a$ and maximum size $b$. Take $S \subseteq \bigcup_{i=1}^m B_i$ such that:*

$$\frac{|S|}{\sum_{i=1}^m |B_i|} \geq 1 - \delta$$

*Let:*

$$S' := \left\{ i \ \middle| \ \frac{|S \cap B_i|}{|B_i|} \geq 1 - \sigma \right\}$$

*Then:*

$$\frac{|S'|}{m} \geq 1 - \frac{(\sigma - \delta)a}{(\sigma - \delta)a + \delta b}$$

*Proof.* Let $\bar{B} := (B_1, \ldots, B_m)$. We look for an upper bound on:

$$\gamma_{\bar{B}, S} := 1 - \frac{|S'|}{m}$$

To do this, take any $(\bar{B}, S)$ with proportion at least $1 - \delta$ which minimises $\gamma_{\bar{B}, S}$. By performing a series of transformations, we show that we can assume that $(\bar{B}, S)$ is of a particular form.

- First, ensuring that $S \cap B_i = B_i$ for each $i \in S'$ does not increase $\gamma_{\bar{B}, S}$ and does not decrease the proportion $\frac{|S|}{\sum_{i=1}^m |B_i|}$.

- We can also make sure that $|B_i| = b$ for each $i \in S'$.

- Similarly, we can ensure that:
$$\frac{|S \cap B_i|}{|B_i|} = \lfloor 1 - \sigma \rfloor$$
  for each $i \notin S'$.

- Finally, we can make sure that $|B_i| = a$ for each $i \notin S'$.

Now, using this nice form for $(\bar{B}, S)$, we can compute:

$$1 - \delta \leq \frac{|S|}{\sum_{i=1}^m |B_i|} \leq \frac{\gamma_{\bar{B}, S} b + (1 - \gamma_{\bar{B}, S})(1 - \sigma)a}{\gamma_{\bar{B}, S} b + (1 - \gamma_{\bar{B}, S})a}$$

Rearranging we get:

$$\frac{|S'|}{m} = 1 - \gamma_{\bar{B}, S} \geq 1 - \frac{(\sigma - \delta)a}{(\sigma - \delta)a + \delta b}$$

as required. $\qquad\square$

*Proof of Lemma C.6.* We proceed differently depending on which of the non-sparse conditions are satisfied.

We begin with the **Density** condition. Say $p$ converges to some $\tilde{p} > 0$. Fix $\gamma, \theta > 0$. Now choose $\epsilon, \delta' > 0$ small enough. We will decide how small later, but we need at least $\epsilon < \tilde{p}$.

By Lemma C.5 (1) there is $N$ such that for all $n \geq N$, with probability at least $1 - \theta$, for all tuples $\bar{u}$ which satisfy $\phi(\bar{u})$ we have that:

$$n(\tilde{p} - \epsilon) < d(u_i) < n(\tilde{p} + \epsilon)$$

Condition on the event that this holds.

Take any $\delta < \delta'$ and $S \subseteq \{(\bar{u}, v) \mid \phi(\bar{u}) \wedge u_i E v\}$ such that:

$$\frac{|S|}{|\{(\bar{u}, v) \mid \phi(\bar{u}) \wedge u_i E v\}|} \geq 1 - \delta$$

Let:

$$S' := \left\{ \bar{u} \;\middle|\; \phi(\bar{u}) \text{ and } \frac{|\{v \mid (\bar{u}, v) \in S\}|}{d(u_i)} \geq 1 - (\delta + \delta^2) \right\}$$

Applying Lemma C.7 with $\sigma = \delta + \delta^2$, $a = n(\tilde{p} - \epsilon)$ and $n = n(\tilde{p} + \epsilon)$, we get that:

$$\frac{|S'|}{|\{\bar{u} \mid \phi(\bar{u}) \text{ and } d(u_i) > 0\}|} \geq 1 - \frac{(\delta + \delta^2 - \delta)n(\tilde{p} - \epsilon)}{(\delta + \delta^2 - \delta)n(\tilde{p} - \epsilon) + \delta n(\tilde{p} + \epsilon)}$$

$$= 1 - \frac{\delta(\tilde{p} - \epsilon)}{\delta(\tilde{p} - \epsilon) + (\tilde{p} + \epsilon)}$$

By making $\epsilon$ and $\delta'$ small enough, we can make this greater than $1 - \gamma$. This completes the proof for the dense case.

We now give the proof for the case of **Root growth**. Let $p(n) = Kn^{-\beta}$. Fix $\gamma, \theta > 0$. Choose $\delta' > 0$ small enough. Also choose $R_1 \in (0, 1)$ and $R_2 > 1$.

By Lemma C.5 (2) there is $N \in \mathbb{N}$ such that for all $n \geq N$, with probability at least $1 - \theta$, for all tuples $\bar{u}$ which satisfy $\phi(\bar{u})$ we have that:

$$R_1 K n^{1-\beta} < d(u_i) < R_2 K n^{1-\beta}$$

Proceeding analogously to the dense case, we have that for all $\delta < \delta'$:

$$\frac{|S'|}{|\{\bar{u} \mid \phi(\bar{u}) \text{ and } d(u_i) > 0\}|} \geq 1 - \frac{\delta R_1 K n^{1-\beta}}{\delta n R_1 K n^{1-\beta} + R_2 K n^{1-\beta}}$$

$$= 1 - \frac{\delta R_1}{\delta R_1 + R_2}$$

By making $\delta'$ and $R_2 - R_1$ small enough, we can make this greater than $1 - \gamma$. This completes the proof for the root growth case.

Finally, we give the argument for the case of **Logarithmic growth**. Let $p(n) = K\frac{\log(n)}{n}$. Fix $\gamma, \theta > 0$. Choose $\zeta, \delta' > 0$ small enough. Also choose $R_1 \in (0, 1)$ and $R_2 > 1$.

By Lemma C.5 (3), there is $R_3 > 0$ and $N \in \mathbb{N}$ such that for all $n \geq N$, with probability at least $1 - \theta$ when drawing graphs from $\mathrm{ER}(n, p(n))$, for at least $1 - \delta$ of the tuples $\bar{u}$ which satisfy $\phi(\bar{u})$ we have that:

$$R_1 K \log(n) < d(u_i) < R_2 K \log(n)$$

and moreover for all tuples $\bar{u}$ which satisfy $\phi(\bar{u})$ we have that:

$$d(u_i) < R_3 K \log(n)$$

Now, let:

$$Q := \{\bar{u} \mid R_1 K \log(n) < d(u_i) < R_2 K \log(n)\}$$

For $n$ large enough we have both:

$$\frac{|Q|}{|\{\bar{u} \mid \phi(\bar{u}) \text{ and } d(u_i) > 0\}|} \geq 1 - \zeta$$

and (using the fact that outside of $Q$ all nodes $u_i$ have degree at most $R_3 K \log(n)$):

$$\frac{|\{(\bar{u}, v) \mid \phi(\bar{u}) \wedge u_i E v \text{ and } \bar{u} \in Q\}|}{|\{(\bar{u}, v) \mid \phi(\bar{u}) \wedge u_i E v\}|} \geq 1 - \zeta$$

Since we have control over $\zeta$, it suffices to restrict attention to:

$$S \subseteq \{(\bar{u}, v) \mid \phi(\bar{u}) \wedge u_i E v \text{ and } \bar{u} \in Q\}$$

Then, analogously to the dense case, we have that for the worst case, for every $\delta < \delta'$:

$$\frac{|S'|}{|\{\bar{u} \mid \phi(\bar{u}) \text{ and } d(u_i) > 0\}|} \geq 1 - \frac{\delta R_1}{\delta R_1 + R_2}$$

By making $\delta'$ and $R_2 - R_1$ small enough, we can make this greater than $1 - \gamma$.

This completes the proof of Lemma C.6 for the logarithmic growth case. $\qquad \square$

The following lemma will be needed only in the analysis of random walk embeddings. It gives the asymptotic behaviour of the random walk embeddings in the non-sparse cases. We see that the embeddings are almost surely zero.

**Lemma C.8.** *Let* $p\colon \mathbb{N} \to \mathbb{N}$ *satisfy density, root growth or logarithmic growth. Then for every* $k \in \mathbb{N}$ *and every* $\epsilon, \delta, \theta > 0$ *there is* $N \in \mathbb{N}$ *such that for all* $n \geq N$ *with probability at least* $1 - \theta$, *the proportion of nodes* $v$ *such that:*

$$\|rw_k(v)\| < \epsilon$$

*is at least* $1 - \delta$.

*Proof.* We start with the **Dense case**. Let $p$ converge to $\tilde{p} > 0$. Take $\epsilon, \delta, \theta > 0$. By Hoeffding's Inequality (Theorem B.4) and taking a union bound, there is $N$ such that for all $n \geq N$ with probability at least $1 - \theta$ we have that:

$$|d(v) - \tilde{p}n| < \epsilon n$$

Condition on this event.

For any node $v$ the number of length-$k$ walks from $v$ is at least:

$$((\tilde{p} - \epsilon)n)^k$$

By removing the last node of the walk, the number of length-$k$ walks from $v$ to itself is at most the number of length-$(k-1)$ walks from $v$.

The number of length-$(k-1)$ walks from $v$ is at most:

$$((\tilde{p} + \epsilon)n)^{k-1}$$

Thus the proportion of length-$k$ walks from $v$ which return to $v$ is at most:

$$\frac{((\tilde{p} + \epsilon)n)^{k-1}}{((\tilde{p} - \epsilon)n)^k} = \frac{(\tilde{p} + \epsilon)^{k-1}}{(\tilde{p} - \epsilon)^k} n^{-1}$$

This tends to 0 an $n$ tends to infinity.

We now argue this for the **Root growth case**. Let $p = Kn^{-\beta}$.

As in the proof of Lemma C.6, by Chernoff's Inequality (Theorem B.3) and taking a union bound, there are $0 < R_1 < 1 < R_2$ and $N$ such that for all $n \geq N$ with probability at least $1 - \theta$ we have that for all $v$:

$$R_1 Kn^{1-\beta} < d(v) < R_2 Kn^{1-\beta}$$

Then, as in the dense case, the proportion of length-$k$ walks from $v$ which return to $v$ is at most:

$$\frac{\left(R_2 Kn^{1-\beta}\right)^{k-1}}{\left(R_1 Kn^{1-\beta}\right)^k} = \frac{R_2^{k-1}}{R_1^k K} n^{\beta-1}$$

which tends to 0 as $n$ tends to infinity.

Finally, we argue this for the **Logarithmic growth case**. Let $p = K \log(n)$. Take $\epsilon, \delta, \theta > 0$.

As in the proof of Lemma C.6, by Chernoff's Inequality (Theorem B.3), there are $0 < R_1 < 1 < R_2$ and $N$ such that for all $n \geq N_1$ with probability at least $1 - \theta$ we have that for at least $1 - \delta$ proportion of nodes $v$:

$$R_1 K \log(n) < d(v) < R_2 K \log(n) \tag{$\bigstar$}$$

Moreover, by Chernoff's Inequality and a union bound, there is $R_3 > 0$ such that for all $n \geq N_2$ with probability at least $1 - \theta$ we have that for all $v$:

$$d(v) < R_3 K \log(n)$$

Let $N := \max(N_1, N_2)$. Take $n \geq N$. We will condition on the event that the above inequalities hold.

Take any node $v$ such that equation ($\bigstar$) holds. Then, the number of length-$k$ walks from $v$ is at least:

$$R_1 K \log(n)((1 - \delta)(R_1 K \log(n)))^{k-1}$$

The number of length-$k$ walks from $v$ to itself is at most:

$$(R_3 K \log(n))^{k-1}$$

Thus the proportion of length-$k$ walks from $v$ which return to $v$ is at most:

$$\frac{(R_3 K \log(n))^{k-1}}{R_1 K \log(n)((1 - \delta)(R_1 K \log(n)))^{k-1}} = \frac{R_3^{k-1}}{R_1^k (1 - \delta)^{k-1}} \cdot \frac{1}{\log(n)}$$

This tends to 0 as $n$ tends to infinity. $\qquad\square$

Finally, we need a simple lemma about division, which has a straightforward proof:

**Lemma C.9.** *Let $x, y, z, w \in \mathbb{R}$ and $\zeta, \xi, \nu, \Omega > 0$ with $\nu > \xi$ be such that $|x - y| < \zeta$ and $|z - w| < \xi$ while $|x|, |z| < \Omega$ and $z > \nu$. Then:*

$$\left| \frac{x}{z} - \frac{y}{w} \right| < \frac{\Omega(\zeta + \xi)}{\nu(\nu - \xi)}$$

### C.2   Proving the inductive invariant for the non-sparse cases, and proving the convergence theorem for these cases

With the growth lemmas for the non-sparse cases in hand, we return to presenting the main proof of Theorem C.1, the convergence theorem for non-sparse ER distributions.

Throughout the following subsection, for notational convenience we allow empty tuples.

For a tuple $\bar{u}$ in a graph let $\mathrm{GrTp}(\bar{u})$ be the graph type of $\bar{u}$. For $k \in \mathbb{N}$ let $\mathsf{GrTp}_k$ be the set of such types with $k$ free variables. For any $t \in \mathsf{GrTp}_k$ let:

$$\mathsf{Ext}(t) := \{t'(\bar{u}, v) \in \mathsf{GrTp}_{k+1} \mid t'(\bar{u}, v) \vDash t(\bar{u})\}$$

For any $u_i$ free in $t$ let:

$$\mathsf{Ext}_{u_i}(t) := \{t'(\bar{u}, v) \in \mathsf{GrTp}_{k+1} \mid t'(\bar{u}, v) \vDash t(\bar{u}) \wedge u_i E v\}$$

We now are ready to begin the proof of Theorem 5.1 for the non-sparse cases. This will involve *defining controllers* and *proving that they satisfy the inductive invariant*.

Let $\mathbb{F}$ be the probability distribution of the node features, which, for the sake of convenience, we assume to have domain $[0, 1]^d$. The domain $[0, 1]^d$ can be replaced by the more general domain $[a, b]^d$ in the results and proofs without further modification. But we note that the domain $[0, 1]^d$ is already sufficient for the results to hold in the general case. Indeed, suppose $\tau(\bar{x})$ is a term which we apply to a graph distribution $\mathcal{D}$ with features in $[a, b]^d$. Modify this distribution to $\mathcal{D}'$ by applying the function $\bar{z} \mapsto (\bar{z} - a)/(b - a)$ to the features. This is now a distribution with features in $[0, 1]^d$. Modify $\tau$ to $\tau'$ by replacing each $\mathrm{H}(x)$ by $F(\mathrm{H}(x))$, where $F$ is the function $\bar{z} \mapsto (b - a)\bar{z} + a$. Then evaluating $\tau'$ on $\mathcal{D}'$ is equivalent to evaluating $\tau$ on $\mathcal{D}$.

Note that in each case the probability function $p(n)$ converges to some $\tilde{p}$ (which is 0 in the root growth and logarithmic growth cases).

Recall from the body of the paper that a $(\bar{u}, d)$ *feature-type controller* is a Lipschitz function taking as input pairs consisting of a $d$-dimensional real vector and a $\bar{u}$ graph type. Recall from Theorem 5.3 that for each term $\pi$, we need to define a controller $e_\pi$ that approximates it. We first give the construction of $e_\pi$ and then recall and verify what it means for $e_\pi$ to approximate $\pi$.

Take $\bar{a} \in ([0, 1]^d)^k$ and $t \in \mathsf{GrTp}_k$, where $k$ is the number of free variables in $\pi$.

When $\pi = \mathrm{H}(u)$ define:

$$e_\pi(\bar{a}, t) := \bar{a}$$

When $\pi = \boldsymbol{c}$, a constant, define:
$$e_\pi(\bar{\boldsymbol{a}}, t) := \boldsymbol{c}$$

When $\pi = \mathrm{rw}(u)$ define:
$$e_\pi(\bar{\boldsymbol{a}}, t) := \boldsymbol{0}$$

When $\pi = f(\rho_1, \dots, \rho_r)$ define:
$$e_\pi(\bar{\boldsymbol{a}}, t) := f(e_{\rho_1}(\bar{\boldsymbol{a}}, t), \dots, e_{\rho_r}(\bar{\boldsymbol{a}}, t))$$

We start with the construction for the global weighted mean. For any $t(\bar{u}) \in \mathsf{GrTp}_k$ and $t'(\bar{u}, v) \in \mathsf{Ext}(t)$, define $\alpha(t, t')$ as follows. As an extension type, $t'$ specifies some number, say $r$, of edges between the nodes $\bar{u}$ and the new node $v$. Define:
$$\alpha(t, t') := \tilde{p}^r(1 - \tilde{p})^{k-r}$$
Note that in both the root growth and logarithmic growth cases, $\alpha(t, t')$ is non-zero precisely when $t'$ specifies no edges between $\bar{u}$ and $v$. These *relative atomic type weights* will play a key role in the construction.

Now consider a term $\pi = \sum_v \rho(\bar{u}, v) \star h(\eta(\bar{u}, v))$.

Recall that the semantics are:
$$\frac{\sum_{v \in G} [\![\rho(\bar{u}, v)]\!]_G h([\![\eta(\bar{u}, v)]\!]_G)}{\sum_{v \in G} h([\![\eta(\bar{u}, v)]\!]_G)}$$

Define:
$$e_\pi(\bar{\boldsymbol{a}}, t) := \frac{f_\pi(\bar{\boldsymbol{a}}, t)}{g_\pi(\bar{\boldsymbol{a}}, t)}$$

where:
$$f_\pi(\bar{\boldsymbol{a}}, t) := \sum_{t' \in \mathsf{Ext}(t)} \alpha(t, t') \mathop{\mathbb{E}}_{\boldsymbol{b} \sim \mathbb{F}} [e_\rho(\bar{\boldsymbol{a}}, \boldsymbol{b}, t') h(e_\eta(\bar{\boldsymbol{a}}, \boldsymbol{b}, t'))]$$

and:
$$g_\pi(\bar{\boldsymbol{a}}, t) := \sum_{t' \in \mathsf{Ext}(t)} \alpha(t, t') \mathop{\mathbb{E}}_{\boldsymbol{b} \sim \mathbb{F}} [h(e_\eta(\bar{\boldsymbol{a}}, \boldsymbol{b}, t'))]$$

Note that when $\tilde{p} = 0$ (as in the root growth and logarithmic growth cases), we have that:
$$\alpha(t, t') = \begin{cases} 1 & \text{if } t' \text{ specifies no edges between } \bar{u} \text{ and } v \\ 0 & \text{otherwise} \end{cases}$$

Therefore the controller becomes, letting $t^\varnothing$ be the type with no edges between $\bar{u}$ and $v$:
$$e_\pi(\bar{\boldsymbol{a}}, t) = \frac{\mathbb{E}_{\boldsymbol{b} \sim \mathbb{F}} [e_\rho(\bar{\boldsymbol{a}}, \boldsymbol{b}, t^\varnothing) h(e_\eta(\bar{\boldsymbol{a}}, \boldsymbol{b}, t^\varnothing))]}{\mathbb{E}_{\boldsymbol{b} \sim \mathbb{F}} [h(e_\eta(\bar{\boldsymbol{a}}, \boldsymbol{b}, t^\varnothing))]}$$

For the local weighted mean case, given $t(\bar{u}) \in \mathsf{GrTp}_k$ and $t'(\bar{u}, v) \in \mathsf{Ext}_{u_i}(t)$, we define $\alpha_{u_i}(t, t')$ as follows. Let $r$ be the number of edges specified by $t'$ between $\bar{u} \setminus u_i$ and $v$. Define:
$$\alpha_{u_i}(t, t') := \tilde{p}^r(1 - \tilde{p})^{k-r-1}$$

Consider $\pi = \sum_{v \in N(u_i)} \rho(\bar{u}, v) \star h(\eta(\bar{u}, v))$. Define:
$$e_\pi(\bar{\boldsymbol{a}}, t) := \frac{f_\pi(\bar{\boldsymbol{a}}, t)}{g_\pi(\bar{\boldsymbol{a}}, t)}$$

where:
$$f_\pi(\bar{\boldsymbol{a}}, t) := \sum_{t' \in \mathsf{Ext}_{u_i}(t)} \alpha_{u_i}(t, t') \mathop{\mathbb{E}}_{\boldsymbol{b} \sim \mathbb{F}} [e_\rho(\bar{\boldsymbol{a}}, \boldsymbol{b}, t') h(e_\eta(\bar{\boldsymbol{a}}, \boldsymbol{b}, t'))]$$

and:
$$g_\pi(\bar{\boldsymbol{a}}, t) := \sum_{t' \in \mathsf{Ext}_{u_i}(t)} \alpha_{u_i}(t, t') \mathop{\mathbb{E}}_{\boldsymbol{b} \sim \mathbb{F}} [h(e_\eta(\bar{\boldsymbol{a}}, \boldsymbol{b}, t'))]$$

With the controllers $e_\pi$ defined, we now prove that they satisfy Theorem (Theorem 5.3). We state the strengthened version of this result here using the notation of $\wedge$-decriptions from Definition C.3:

**Lemma C.10.** *For every $\epsilon, \delta, \theta > 0$ and $\wedge$-description $\psi(\bar{u})$, there is $N \in \mathbb{N}$ such that for all $n \geq N$, with probability at least $1 - \theta$ in the space of graphs of size $n$, out of all the tuples $\bar{u}$ such that $\psi(\bar{u})$, at least $1 - \delta$ satisfy:*

$$\|e_\pi(\mathrm{H}(\bar{u}), \mathrm{GrTp}(\bar{u})) - [\![\pi(\bar{u})]\!]\| < \epsilon$$

*Proof.* Naturally, we prove the lemma by induction.

- **Base cases** $\mathrm{H}(v)$ **and constant** $c$. These are immediate from the definition.

- **Base case** $\mathrm{rw}(v)$. This follows by Lemma C.8.

- **Induction step for Lipschitz functions** $f(\rho_1, \ldots, \rho_r)$.

  Take $\epsilon, \delta, \theta > 0$ and $\wedge$-description $\psi(\bar{u})$. By the induction hypothesis there is $N \in \mathbb{N}$ such that for all $n \geq N$ and for every $i \leq r$, with probability at least $1 - \theta$, we have that out of all the tuples $\bar{u}$ such that $\psi(\bar{u})$, at least $1 - \delta$ satisfy:

  $$\|e_{\rho_i}(\mathrm{H}(\bar{u}), \mathrm{GrTp}(\bar{u})) - [\![\rho_i(\bar{u})]\!]\| < \epsilon$$

  Hence, with probability at least $1 - r\theta$, out of all the tuples $\bar{u}$ such that $\psi(\bar{u})$, at least $1 - r\delta$ satisfy this for every $i \leq r$. Condition on this event.

  Take any such tuple $\bar{u}$. Then by Lipschitzness of $f$ we have that the normed distance between:

  $$e_\pi(\mathrm{H}(\bar{u}), \mathrm{GrTp}(\bar{u})) = f(e_{\rho_1}(\mathrm{H}(\bar{u}), \mathrm{GrTp}(\bar{u})), \ldots, e_{\rho_r}(\mathrm{H}(\bar{u}), \mathrm{GrTp}(\bar{u})))$$

  and:

  $$[\![\pi(\bar{u})]\!] = f([\![\rho_1(\bar{u})]\!], \ldots, [\![\rho_r(\bar{u})]\!])$$

  is at most $L_f \epsilon$, where $L_f$ is the Lipschitz constant of $f$. Since $L_f$ is a constant, this case follows.

- **Inductive step for** $\sum_v \rho(\bar{u}, v) \star h(\eta(\bar{u}, v))$.

  Take $\epsilon, \delta, \theta > 0$. Take a $\wedge$-description $\psi(\bar{u})$.

  By the induction hypothesis, there is $N_1$ such that for all $n \geq N_1$, with probability at least $1 - \theta$, out of all the tuples $(\bar{u}, v)$ which satisfy $\psi(\bar{u})$, at least $1 - \delta$ satisfy:

  $$\|e_\rho(\mathrm{H}(\bar{u}, v), \mathrm{GrTp}(\bar{u}, v)) - [\![\rho(\bar{u}, v)]\!]\| < \epsilon \tag{$\dagger_\rho$}$$

  and:

  $$\|e_\eta(\mathrm{H}(\bar{u}, v), \mathrm{GrTp}(\bar{u}, v)) - [\![\eta(\bar{u}, v)]\!]\| < \epsilon \tag{$\dagger_\eta$}$$

  Take $\gamma > 0$. We will choose how small $\gamma$ is later.

  By Lemma C.6 there is $N_\gamma > N_1$ such that for all $n \geq N_\gamma$, with probability at least $1 - \theta$, out of all the tuples $\bar{u}$ such that $\psi(\bar{u})$, at least $1 - \gamma$, at least $1 - (\delta + \delta^2)$ of the $v$'s satisfy equation $(\dagger_\rho)$ and equation $(\dagger_\eta)$.

  Now consider $t(\bar{u}) \in \mathsf{GrTp}_k$ and take any $t'(\bar{u}, v) \in \mathsf{Ext}(t)$. For any tuple $\bar{u}$ of nodes such that $\mathrm{GrTp}(\bar{u}) = t$, define:

  $$[\![t'(\bar{u})]\!] := \{v \mid t'(\bar{u}, v)\}$$

  Since $p(n)$ converges to $\tilde{p}$ and $\mathsf{Ext}(t)$ is finite, there is $N_3$ such that for all $n \geq N_3$ with probability at least $1 - \theta$ we have that for every pair $(t, t')$ and tuple $\bar{u}$ of nodes such that $t(\bar{u})$:

  $$\left| \frac{|[\![t'(\bar{u})]\!]|}{n} - \alpha(t, t') \right| < \frac{1}{2^{k+1}} \epsilon \tag{$*$}$$

  Next, for any tuple $\bar{u}$ consider the function:

  $$f_\pi^\circ(\bar{u}, v) := e_\rho(H_G(\bar{u}, v), \mathrm{GrTp}(\bar{u}, v)) h(e_\eta(H_G(\bar{u}, v), \mathrm{GrTp}(\bar{u}, v)))$$

  Note that the function:

  $$(\bar{a}, \boldsymbol{b}, t') \mapsto e_\rho(\bar{a}, \boldsymbol{b}, t') h(e_\eta(\bar{a}, \boldsymbol{b}, t'))$$

  is bounded. Let $\Lambda$ be the diameter of its range. This also bounds $f_\pi^\circ$.

We will now apply Hoeffding's Inequality to $\sum_{v|t'(\bar{u},v)} f_\pi^\circ(\bar{u}, v)$. Note that the number of summands is $|[\![ t'(\bar{u}) ]\!]|$ and the summands are bounded by $\Lambda$. Furthermore, the summands are independent and have expected value:

$$\underset{\boldsymbol{b} \sim \mathbb{F}}{\mathbb{E}} [e_\rho(H_G(\bar{u}, \boldsymbol{b}), t') h(e_\eta(H_G(\bar{u}, \boldsymbol{b}), t'))]$$

Hence, by Hoeffding's Inequality (Theorem B.4) for any $\bar{u}$ the probability that:

$$\left\| \frac{1}{n} \sum_{v|t'(\bar{u},v)} f_\pi^\circ(\bar{u}, v) - \frac{|[\![ t'(\bar{u}) ]\!]|}{n} \underset{\boldsymbol{b} \sim \mathbb{F}}{\mathbb{E}} [e_\rho(H_G(\bar{u}, \boldsymbol{b}), t') h(e_\eta(H_G(\bar{u}, \boldsymbol{b}), t'))] \right\| \geq \frac{1}{2^{k+1}} \epsilon \quad (\heartsuit)$$

is at most:

$$2d \exp\left( -\frac{\epsilon^2 |[\![ t'(\bar{u}) ]\!]|}{2\Lambda^2} \right)$$

By equation ($*$) there is $N_4$ such that for all $n \geq N_4$, with probability at least $1 - \theta$ for all $t'$ such that $\alpha(t, t') > 0$ we have that for every tuple $\bar{u}$:

$$2d \exp\left( -\frac{\epsilon^2 |[\![ t'(\bar{u}) ]\!]|}{2\Lambda^2} \right) < \theta\delta \quad (\clubsuit)$$

Let $B_n$ be the proportion, out of all tuples $\bar{u}$ for which $\psi(\bar{u})$ holds[4], such that:

$$\left\| \frac{1}{n} \sum_{v|t'(\bar{u},v)} f_\pi^\circ(\bar{u}, v) - \frac{|[\![ t'(\bar{u}) ]\!]|}{n} \underset{\boldsymbol{b} \sim \mathbb{F}}{\mathbb{E}} [e_\rho(H_G(\bar{u}, \boldsymbol{b}), t') h(e_\eta(H_G(\bar{u}, \boldsymbol{b}), t'))] \right\| \geq \frac{1}{2^{k+1}} \epsilon$$

Note that the property above is exactly the event whose probability is bounded in equation ($\heartsuit$). We can express $B_n$ as the mean of a set of indicator variables, one for each tuple $\bar{u}$ such that $\psi(\bar{u})$ holds. Each indicator variable has expected value at most $\theta\delta$ by equation ($\heartsuit$) and equation ($\clubsuit$).

Then by linearity of expectation, $\mathbb{E}[B_n] \leq \theta\delta$ for all $n \geq N_4$, and hence by Markov's Inequality (Theorem B.1):

$$\mathbb{P}(B_n \geq \delta) \leq \frac{\theta\delta}{\delta} = \theta$$

Therefore, for all $n \geq \max(N_3, N_4)$, with probability at least $1 - \theta$ for at least $1 - \delta$ of the tuples $\bar{u}$ for which $\psi(\bar{u})$ holds we have that:

$$\left\| \frac{1}{n} \sum_{v|t'(\bar{u},v)} f_\pi^\circ(\bar{u}, v) - \frac{|[\![ t'(\bar{u}) ]\!]|}{n} \underset{\boldsymbol{b} \sim \mathbb{F}}{\mathbb{E}} [e_\rho(H_G(\bar{u}, \boldsymbol{b}), t') h(e_\eta(H_G(\bar{u}, \boldsymbol{b}), t'))] \right\| < \frac{1}{2^{k+1}} \epsilon$$

and therefore, by equation ($*$) and the definition of $f_\pi$:

$$\left\| \frac{1}{n} \sum_v f_\pi^\circ(\bar{u}, v) - f_\pi(H_G(\bar{u}), \mathrm{GrTp}(\bar{u})) \right\| < \epsilon \quad (\triangle_f)$$

Similarly, for any tuple $\bar{u}$ consider the function:

$$g_\pi^\circ(\bar{u}, v) := h(e_\eta(H_G(\bar{u}, v), \mathrm{GrTp}(\bar{u}, v)))$$

As above there is $N_5 \geq N_3$ such that for all $n \geq N_5$ with probability at least $1 - \theta$ we have that the proportion out of tuples $\bar{u}$ for which $\psi(\bar{u})$ holds such that:

$$\left\| \frac{1}{n} \sum_v g_\pi^\circ(\bar{u}, v) - g_\pi(H_G(\bar{u}), \mathrm{GrTp}(\bar{u})) \right\| < \epsilon \quad (\triangle_g)$$

is at least $1 - \delta$.

---

[4]Recall that $\psi$ is the $\wedge$-description on which we condition the tuples $\bar{u}$ in the inductive invariant.

Take $n \geq \max(N_1, N_\gamma, N_3, N_4, N_5)$. For such an $n$, these events above hold with probability at least $1 - 5\theta$. So from now on we will condition on the event that they hold.

Take any such tuple $\bar{u}$. Let $t := \mathrm{GrTp}(\bar{u})$. It suffices to show, using the definition of the interpretation of the weighted mean operator, that:

$$\left\| \frac{\sum_v [\![\rho(\bar{u}, v)]\!] h([\![\eta(\bar{u}, v)]\!])}{\sum_v h([\![\eta(\bar{u}, v)]\!])} - \frac{f_\pi(H_G(\bar{u}), t)}{g_\pi(H_G(\bar{u}), t)} \right\| < \iota(\epsilon, \delta, \gamma)$$

for $\iota$ which we can make arbitrarily small by choosing $\epsilon, \delta, \gamma$ small enough.

By Lemma C.9 it suffices to find $\zeta(\epsilon, \delta, \gamma), \xi(\epsilon, \delta, \gamma) > 0$ which we can make arbitrarily small and constants $\nu, \Omega > 0$ such that:

$$\left\| \frac{1}{n} \sum_v [\![\rho(\bar{u}, v)]\!] h([\![\eta(\bar{u}, v)]\!]) - f_\pi(H_G(\bar{u}), t) \right\| < \zeta(\epsilon, \delta, \gamma) \tag{1}$$

$$\left\| \frac{1}{n} \sum_v h([\![\eta(\bar{u}, v)]\!]) - g_\pi(H_G(\bar{u}), t) \right\| < \xi(\epsilon, \delta, \gamma) \tag{2}$$

$$\left\| \sum_v [\![\rho(\bar{u}, v)]\!] h([\![\eta(\bar{u}, v)]\!]) \right\| < \Omega n \tag{3}$$

$$\left\| \sum_v h([\![\eta(\bar{u}, v)]\!]) \right\| < \Omega n \tag{4}$$

$$\forall i \leq d: \ \left[ \sum_v h([\![\eta(\bar{u}, v)]\!]) \right]_i > \nu n \tag{5}$$

For equation (3) and equation (4) we can use the fact that $[\![\rho(\bar{u}, v)]\!]$ and $[\![\eta(\bar{u}, v)]\!]$ are bounded and that $h$ is Lipschitz on bounded domains. For equation (5) we use the fact that $[\![\eta(\bar{u}, v)]\!]$ is bounded and that the codomain of $h$ is $(0, \infty)^d$.

The proofs of equation (1) and equation (2) are very similar. We prove equation (2) since it is slightly notationally lighter. Let:

$$g_\pi^*(\bar{u}) := \frac{1}{n} \sum_v h(e_\eta(H_G(\bar{u}, v), t))$$

Let $\kappa$ be a bound on the norms of $h(e_\eta(\bar{a}, \boldsymbol{b}, t'))$ and $h([\![\eta(\bar{u}, v)]\!])$. By equation $(\dagger_\eta)$ we have that:

$$\left\| \frac{1}{n} \sum_v h([\![\eta(\bar{u}, v)]\!]) - g_\pi^*(\bar{u}) \right\| < \left(1 - (\delta + \delta^2)\right) \epsilon + (\delta + \delta^2) 2\kappa^2$$

We can make the right-hand-side as small as we like by taking $\epsilon$ and $\delta$ sufficiently small.

Now note that:

$$g_\pi^*(\bar{u}) = \frac{1}{n} \sum_v g_\pi^\circ(\bar{u}, v)$$

Hence by equation $(\triangle_g)$ we have that:

$$\|g_\pi^*(\bar{u}) - g_\pi(H_G(\bar{u}), t)\| < \epsilon$$

Therefore:

$$\left\| g_\pi(H_G(\bar{u}), t) - \frac{1}{n} \sum_v h([\![\eta(\bar{u}, v)]\!]) \right\| < \left(1 - (\delta + \delta^2)\right) \epsilon + (\delta + \delta^2) 2\kappa^2 + \epsilon$$

which we can make as small as we like by taking $\epsilon$ and $\delta$ small enough.

- **Inductive step for** $\sum_{v\in\mathcal{N}(u_i)}\rho(\bar{u},v)\star h(\eta(\bar{u},v))$.

  We proceed similarly to the global weighted mean case, this time making use of the conditioning $\wedge$-description in the inductive invariant. Indeed, notice that when we apply the inductive invariant below, we add $u_i E v$ to our condition.

  Take $\epsilon,\delta,\theta>0$. Take a $\wedge$-description $\psi(\bar{u})$.

  By the induction hypothesis, there is $N_1$ such that for all $n\geq N_1$, with probability at least $1-\theta$, out of all the tuples $(\bar{u},v)$ which satisfy $\psi(\bar{u})\wedge u_i E v$, at least $1-\delta$ satisfy:
  $$\|e_\rho(H_G(\bar{u},v),\mathrm{GrTp}(\bar{u},v))-[\![\rho(\bar{u},v)]\!]\|<\epsilon \tag{$\dagger_\rho^{loc}$}$$
  and:
  $$\|e_\eta(H_G(\bar{u},v),\mathrm{GrTp}(\bar{u},v))-[\![\eta(\bar{u},v)]\!]\|<\epsilon \tag{$\dagger_\eta^{loc}$}$$

  Take $\gamma>0$. We will choose how small $\gamma$ is later.

  By Lemma C.6 there is $N_\gamma>N_1$ such that for all $n\geq N_\gamma$, with probability at least $1-\theta$ the following event happens. Out of all the tuples $\bar{u}$ such that $\psi(\bar{u})\wedge\exists v\colon u_i E v$, a proportion of at least $1-\gamma$ of them satisfy the following. At least $1-(\delta+\delta^2)$ of the nodes $v$ for which $\psi(\bar{u})\wedge u_i E v$ holds also satisfy equation ($\dagger_\rho^{loc}$) and equation ($\dagger_\eta^{loc}$).

  Now consider $t(\bar{u})\in\mathsf{GrTp}_k$ and take any $t'(\bar{u},v)\in\mathsf{Ext}_{u_i}(t)$. For any tuple $\bar{u}$ of nodes such that $\mathrm{GrTp}(\bar{u})=t$, define:
  $$[\![t'(\bar{u})]\!]:=\{v\in\mathcal{N}(u_i)\mid t'(\bar{u},v)\}$$

  By Lemma C.5 we have that in all the non-sparse cases there is $R>0$ and $N_3$ such that for all $n\geq N_3$ with probability at least $1-\theta$, the proportion of all tuples $\bar{u}$ for which $\psi(\bar{u})$ holds such that:
  $$d(u_i)\geq R\log(n)$$
  is at least $1-\delta$. Then, as before, there is $N_4\geq N_3$ such that for all $n\geq N_4$, with probability at least $1-\theta$, for all pairs $(t,t')$ the proportion of all tuples $\bar{u}$ for which $\psi(\bar{u})$ and $t(\bar{u})$ hold such that:
  $$\left|\frac{|[\![t'(\bar{u})]\!]|}{d(u_i)}-\alpha(t,t')\right|<\frac{1}{2^{k+1}}\epsilon$$

  For any tuple $\bar{u}$ define the function:
  $$f_\pi^\circ(\bar{u},v):=e_\rho(H_G(\bar{u},v),\mathrm{GrTp}(\bar{u},v))h(e_\eta(H_G(\bar{u},v),\mathrm{GrTp}(\bar{u},v)))$$

  Taking $\Lambda$ as before, by Hoeffding's Inequality for any $\bar{u}$ the probability that:
  $$\left\|\frac{1}{d(u_i)}\sum_{v\in\mathcal{N}(u_i)\mid t'(\bar{u},v)}f_\pi^\circ(\bar{u},v)-\frac{|[\![t'(\bar{u})]\!]|}{d(u_i)}\mathop{\mathbb{E}}_{\boldsymbol{b}\sim\mathbb{F}}[e_\rho(H_G(\bar{u},\boldsymbol{b}),t')h(e_\eta(H_G(\bar{u},\boldsymbol{b}),t'))]\right\|$$
  $$\geq\frac{1}{2^{k+1}}\epsilon$$
  is at most:
  $$2d\exp\left(-\frac{\epsilon^2|[\![t'(\bar{u})]\!]|}{2\Lambda^2}\right)$$

  Using a similar argument to the global weighted mean case, there is $N_5$ such that for all $n\geq N_5$, with probability at least $1-\theta$ for all $t'$ such that $\alpha(t,t')>0$ we have that for at least $1-\delta$ of the tuples $\bar{u}$ such that $\psi(\bar{u})$ and $t(\bar{u})$ hold:
  $$2d\exp\left(-\frac{\epsilon^2|[\![t'(\bar{u})]\!]|}{2\Lambda^2}\right)<\theta\delta$$

  We now proceed as in the global weighted mean case, creating a random variable similar to $B_n$ and making using of linearity of expectation. We get that for all $n\geq\max(N_3,N_4,N_5)$, with probability at least $1-\theta$ for at least $1-\delta$ of the tuples $\bar{u}$ for which $\psi(\bar{u})$ holds we have that:
  $$\left\|\frac{1}{n}\sum_v f_\pi^\circ(\bar{u},v)-f_\pi(H_G(\bar{u}),\mathrm{GrTp}(\bar{u}))\right\|<\epsilon \tag{$\triangle_f^{loc}$}$$

Similarly, with $g_\pi^\circ$ defined as above, there is $N_6$ such that for all $n \geq N_6$, with probability at least $1 - \theta$ for at least $1 - \delta$ of the tuples $\bar{u}$ for which $\psi(\bar{u})$ holds we have that:

$$\left\| \frac{1}{n} \sum_v g_\pi^\circ(\bar{u}, v) - g_\pi(H_G(\bar{u}), \mathrm{GrTp}(\bar{u})) \right\| < \epsilon \qquad (\triangle_g^{\mathrm{loc}})$$

With equations $(\dagger_\rho^{loc})$, $(\dagger_\eta^{loc})$, $(\triangle_f^{\mathrm{loc}})$ and $(\triangle_g^{\mathrm{loc}})$ established, we can now proceed as before, making use of Lemma C.9.

This completes the proof of Theorem 5.3. $\qquad\qquad\qquad\square$

**Applying the lemma to prove the theorem.** To prove Theorem C.1, note that $e_\tau$ is only a function of the $\mathrm{GrTp}_0$ (since $\tau$ is a closed term), while $\mathrm{GrTp}_0$ consists of a single type, $\top$, which is satisfied by all graphs. So $e_\tau$ is a constant.

# D   Sparse Erdős-Rényi and Barabási-Albert

We now give the proof for the sparse Erdős-Rényi and Barabási-Albert cases of Theorem 5.1. In addition, we extend the result to the language $\mathrm{AGG}[\mathrm{WMEAN}, \mathrm{GCN}, \mathrm{RW}]$ defined in Appendix A. We state the full result here.

**Theorem D.1.** *Consider $(\mu_n)_{n \in \mathbb{N}}$ sampling a graph $G$ from either of the following models and node features independently from i.i.d. bounded distributions on $d$ features.*

1. *The Erdős-Rényi distribution $\mathrm{ER}(n, p(n))$ where $p$ satisfies **Sparsity**: for some $K > 0$ we have: $p(n) = Kn^{-1}$.*

2. *The Barabási-Albert model $\mathrm{BA}(n, m)$ for any $m \geq 1$.*

*Then for every $\mathrm{AGG}[\mathrm{WMEAN}, \mathrm{GCN}, \mathrm{RW}]$ term converges a.a.s. with respect to $(\mu_n)_{n \in \mathbb{N}}$.*

As discussed in the body, this requires us to analyze neighbourhood isomorphism types, which implicitly quantify over local neighbourhoods, rather than atomic types as in the non-sparse cases.

*Remark D.2.* Formally, a $k$-rooted graph is a tuple $(T, \bar{u})$. At various points below to lighten the notation we drop the '$\bar{u}$' and refer simply to a $k$-rooted graph $T$.

## D.1   Combinatorial tools for the sparse case

As in the dense case, before we turn to analysis of our term language, we prove some results about the underlying random graph model that we can use as tools.

The key combinatorial tool we will use is the following, which states that for any tuple of nodes $\bar{u}$, the proportion of nodes $v$ such that $(\bar{u}, v)$ has any particular neighbourhood type converges.

**Definition D.3.** Let $(T, \bar{w})$ be a $k$-rooted graph and let $(T', \bar{w}, y)$ be a $(k + 1)$-rooted graph. Then $T'$ *extends* $T$ if $T$ is a subgraph of $T'$. Let $\mathrm{Ext}(T)$ be the set of all $(k + 1)$-rooted graphs extending $(T, \bar{w})$.

**Theorem D.4.** *Consider sampling a graph $G$ from either the sparse Erdős-Rényi or Barabási-Albert distributions. Let $(T, \bar{w})$ be a $k$-rooted graph and take $\ell \in \mathbb{N}$. Then for every $(k + 1)$-rooted graph $(T', \bar{w}, y)$ which extends $(T, \bar{w})$ there is $q_{T'|T} \in [0, 1]$ such that for all $\epsilon, \theta > 0$ there is $N \in \mathbb{N}$ such that for all $n \geq N$ with probability at least $1 - \theta$ we have that for all $k$-tuples of nodes $\bar{u}$ such that $\mathcal{N}_\ell^G(\bar{u}) \cong (T, \bar{w})$:*

$$\left| \frac{|\{v \mid \mathcal{N}_\ell^G(\bar{u}, v) \cong (T', \bar{w}, y)\}|}{n} - q_T \right| < \epsilon$$

*Moreover, we have that:*

$$\sum_{T' \in \mathrm{Ext}(T)} q_{T'|T} = 1$$

We refer to the $q_{T'|T}$ as *relative neighborhood weights*. They will play a role in defining the controllers, analogous to that of the relative atomic type weights $\alpha(t, t')$ in the non-sparse case.

Theorem D.4 follows from what is known as "weak local convergence" in the literature. It is essentially a non-parametrised version of Theorem D.4.

**Definition D.5.** A sequence $(\mu_n)_{n \in \mathbb{N}}$ of graph distributions has *weak local convergence* if for every $k$-rooted graph $(T, \bar{w})$ and $\ell \in \mathbb{N}$ there is $q_T \in [0, 1]$ such that for all $\epsilon, \theta > 0$ there is $N \in \mathbb{N}$ such that for all $n \geq N$ with probability at least $1 - \theta$ we have that:

$$\left| \frac{|\{\bar{u} \mid \mathcal{N}_\ell^G(\bar{u}) \cong (T, \bar{w})\}|}{n} - q_T \right| < \epsilon$$

and moreover:

$$\sum_T q_T = 1$$

**Theorem D.6.** *The sparse Erdős-Rényi and Barabási-Albert distributions have weak local convergence.*

*Proof.* See [48, Theorem 2.18] for the sparse Erdős-Rényi distribution and [17, Theorem 4.2.1] for the Barabási-Albert distribution. $\square$

*Proof of Theorem D.4.* Take $T' \in \mathsf{Ext}(T)$. There are two cases.

**Case 1.** There is a path from $y$ to some node $w_i$ of $\bar{w}$ in $T'$.

Set $q_{T'|T} = 0$. Note that in this case, for all tuples $\bar{u}$ such that $\mathcal{N}_\ell^G(\bar{u}) \cong T$ we have that:

$$\{v \mid \mathcal{N}_\ell^G(\bar{u}, v) \cong (T', \bar{w}, y)\} \subseteq \mathcal{N}_\ell^G(u_i)\}$$

Since $\mathcal{N}_\ell^G(u_i)$ is determined by $T$ and is thus of fixed size, the proportion:

$$\left| \frac{|\{v \mid \mathcal{N}_\ell^G(\bar{u}, v) \cong (T', \bar{w}, y)\}|}{n} \right|$$

tends to $0$ as $n$ grows. This completes the proof for Case 1.

**Case 2.** There is no path from $y$ to any node of $\bar{w}$ in $T'$. Then for all tuples $\bar{u}$ such that $\mathcal{N}_\ell^G(\bar{u}) \cong T$ we have that:

$$\{v \mid \mathcal{N}_\ell^G(\bar{u}, v) \cong (T', \bar{w}, y)\} = \{v \mid \mathcal{N}_\ell^G(v) \cong \mathcal{N}_\ell^{T'}(y)\} \setminus \mathcal{N}_\ell^G(\bar{u})$$

Hence:

$$\frac{|\{v \mid \mathcal{N}_\ell^G(v) \cong \mathcal{N}_\ell^{T'}(y)\}|}{n} - \frac{|\mathcal{N}_\ell^G(\bar{u})|}{n} \leq \frac{|\{v \mid \mathcal{N}_\ell^G(\bar{u}, v) \cong (T', \bar{w}, y)\}|}{n}$$

Also:

$$\frac{|\{v \mid \mathcal{N}_\ell^G(\bar{u}, v) \cong (T', \bar{w}, y)\}|}{n} \leq \frac{|\{v \mid \mathcal{N}_\ell^G(v) \cong \mathcal{N}_\ell^{T'}(y)\}|}{n}$$

By Lemma 5.5 (with $k = 1$) we have that:

$$\frac{|\{v \mid \mathcal{N}_\ell^G(v) \cong \mathcal{N}_\ell^{T'}(y)\}|}{n}$$

converges to some limit $q_{T|T'} := q_{\mathcal{N}_\ell^{T'}(y)}$ asymptotically almost surely, while:

$$\frac{|\mathcal{N}_\ell^G(\bar{u})|}{n} = \frac{|T|}{n}$$

converges to $0$. This completes the proof for Case 1.

Finally, to show that $\sum_{T' \in \mathsf{Ext}(T)} q_{T'|T} = 1$, note the set:

$$\{\mathcal{N}_\ell^{T'}(y) \mid (T', \bar{w}, y) \in \mathsf{Ext}(T')\}$$

contains all 1-rooted graphs up to isomorphism which can be the $\ell$-neighbourhood of a node. Therefore, by the "moreover" part of Definition D.5 we have that:

$$\sum_{T' \in \mathsf{Ext}(T)} q_{T'|T} = \sum_{T' \in \mathsf{Ext}(T)} q_{\mathcal{N}_\ell^{T'}(y)} = 1$$

$\square$

## D.2 Proving the inductive invariant for the sparse cases, and proving the convergence theorem for these cases

We now begin the proof of Theorem D.1. Recall from the body that we will do this via Theorem 5.6, which requires us to define some Lipschitz controllers on neighbourhood types that approximate a given term $\pi$ relative to a neighbourhood type.

Throughout the following, it will be convenient to assume for every $k$-rooted graph isomorphism type $(T, \bar{u})$ a canonical ordering nodes $T = \{s_1, \ldots, s_{|T|}\}$ such that:

- the first $k$ nodes in the ordering are $\bar{u}$ and
- for any graph $G$, tuple of nodes $\bar{u}$ and $\ell, \ell' \in \mathbb{N}$ with $\ell' < \ell$ the canonical ordering of $\mathcal{N}_{\ell'}(\bar{u})$ is an initial segment of the canonical ordering of $\mathcal{N}_{\ell}(\bar{u})$.

Again we will use $\mathbb{F}$ for the probability distribution of the node features. For each subterm $\pi$ of $\tau$, its *reach*, denoted $\mathrm{Reach}(\pi)$ is a natural number defined as follows:

$$\mathrm{Reach}(\mathrm{H}(u)) = 0$$
$$\mathrm{Reach}(\boldsymbol{c}) = 0$$
$$\mathrm{Reach}(\mathrm{rw}(v)) = d$$
$$\mathrm{Reach}(f(\rho_1, \ldots, \rho_r)) = \max_{i \leq r} \mathrm{Reach}(\rho_i)$$

$$\mathrm{Reach}\left(\sum_{v \in \mathcal{N}(u)} \rho \star h(\eta)\right) = \max(\mathrm{Reach}(\rho), \mathrm{Reach}(\eta)) + 1$$

$$\mathrm{Reach}\left(\sum_{v} \rho \star h(\eta)\right) = 0$$

Take a subterm $\pi$ of $\tau$. Let $k$ be the number of free variables in $\pi$. We now define the controller at $\pi$ for every $k$-rooted graph $(T, \bar{w})$. The controller will be of the form:

$$e_\pi^T \colon ([0,1]^d)^{|T|} \to \mathbb{R}^d$$

Note that when $|T| = 0$ this is a constant.

Recall from Theorem 5.6 that our goal is to ensure the following correctness property for our controllers $e_\pi^T$.

**Property D.7.** *Let $\pi$ be any subterm of $\tau$ and let $\ell = \mathrm{Reach}(\pi)$. Consider sampling a graph $G$ from either the sparse Erdős-Rényi or Barabási-Albert distributions. For every $k$-rooted graph $(T, \bar{w})$ and $\epsilon, \theta > 0$, there is $N \in \mathbb{N}$ such that for all $n \geq N$ with probability at least $1 - \theta$ we have that for every $k$-tuple of nodes $\bar{u}$ in the graph such that $\mathcal{N}_\ell(\bar{u}) \cong (T, \bar{w})$, taking the canonical ordering $\mathcal{N}_\ell(\bar{u}) = \{s_1, \ldots, s_{|T|}\}$:*

$$\left\| e_\pi^T(H_G(s_1), \ldots, H_G(s_{|T|})) - [\![\pi(\bar{u})]\!] \right\| < \epsilon$$

For notational convenience, we allow each $e_\pi^T$ to take additional arguments as input, which it will ignore.

*Proof of Theorem 5.6.* We give the construction of the controllers in parallel with the proof of correctness.

- **Base case $\pi = \mathrm{H}(u_i)$.**

    Here $\ell = \mathrm{Reach}(\pi) = 0$. Let $e_\pi^T(\bar{a}) := \boldsymbol{a}_i$, the feature value of the $i^{\mathrm{th}}$ node in the tuple. Note that for any $k$-tuple of nodes $\bar{u}$ in the graph we have:

$$e_\pi^T(H_G(s_1), \ldots, H_G(s_{|T|})) = H_G(s_i)$$
$$= [\![\pi(\bar{u})]\!]$$

- **Base case $\pi = \boldsymbol{c}$.**

    Let $e_\pi^T(\bar{a}) := \boldsymbol{c}$.

- **Base case** $\pi = \mathrm{rw}(u_i)$.

  Then $\ell = \mathrm{Reach}(\pi) = d$. Note that the random walk embeddings up to length $d$ are entirely determined by the $d$-neighbourhood. Therefore, given rooted graph $(T, \bar{w})$, there is $\boldsymbol{r}_T$ such that:

  $$\llbracket \mathrm{rw}(u_i) \rrbracket = \boldsymbol{r}_T$$

  for all tuples $\bar{u}$ such that $\mathcal{N}_\ell(\bar{u}) = T$. So set:

  $$e_\pi^T(\bar{\boldsymbol{a}}) = \boldsymbol{r}_T$$

  for any $\bar{\boldsymbol{a}}$.

- **Inductive step for Lipschitz functions** $\pi = F(\rho_1, \ldots, \rho_r)$.

  Note that $\ell = \max_{j \le r} \mathrm{Reach}(\rho_i)$. Take a $k$-rooted graph $(T, \bar{w})$. For each $i \le r$ let $T_j := \mathcal{N}_{\mathrm{Reach}(\rho_j)}^T(\bar{w})$. Define:

  $$e_\pi^T(\bar{\boldsymbol{a}}) := F(e_{\rho_1}^{T_1}(\bar{\boldsymbol{a}}), \ldots, e_{\rho_r}^{T_r}(\bar{\boldsymbol{a}}))$$

  Now take $\epsilon, \theta > 0$. By the induction hypothesis, there is $N$ such that for all $n \ge N$ with probability at least $1 - \theta$ we have that for every $k$-tuple of nodes $\bar{u}$ in the graph, letting $T = \mathcal{N}_\ell(\bar{u})$, for every $j \le r$:

  $$\left\| e_{\rho_j}^{T_j}(H_G(s_1), \ldots, H_G(s_{|T_j|})) - \llbracket \rho_j(\bar{u}) \rrbracket \right\| < \epsilon$$

  Hence, by the Lipschitzness of $F$ we have:

  $$\left\| e_\pi^T(H_G(s_1), \ldots, H_G(s_{|T|})) - \llbracket \pi(\bar{u}) \rrbracket \right\| < L_F \epsilon$$

  where $L_F$ is the Lipschitz constant of $F$.

- **Inductive step for local weighted mean** $\pi = \sum_{v \in \mathcal{N}(u_i)} \rho \star h(\eta)$.

  In this step we use that the value of a local aggregator is determined by the values of the terms it is aggregating in the local neighbourhood.

  Take a $k$-rooted graph $(T, \bar{w})$, where $k$ is the number of free variables in $\pi$. Note that $\ell = \max(\mathrm{Reach}(\rho), \mathrm{Reach}(\eta)) + 1$.

  When $w_i$ has no neighbours in $T$, define:

  $$e_\pi^T(\bar{\boldsymbol{a}}) := \boldsymbol{0}$$

  and note that for any $k$-tuple of nodes $\bar{u}$ in the graph such that $\mathcal{N}_\ell(\bar{u}) = T$ we have:

  $$\llbracket \pi(\bar{u}) \rrbracket = \boldsymbol{0} = e_\pi^T(H_G(s_1), \ldots, H_G(s_{|T|}))$$

  So suppose that $w_i$ has some neighbours in $T$. Enumerate the neighbours as:

  $$\mathcal{N}^T(w_i) = \{y_1, \ldots, y_r\}$$

  Define the Lipschitz function:

  $$\mathrm{WMean}_T(\boldsymbol{a}_1, \ldots, \boldsymbol{a}_r, \boldsymbol{b}_1, \ldots, \boldsymbol{b}_r) := \frac{\sum_{j=0}^r \boldsymbol{a}_j h(\boldsymbol{b}_j)}{\sum_{j=0}^r h(\boldsymbol{b}_j)}$$

  Note that whenever $\mathcal{N}_\ell^G(\bar{u}) \cong (T, \bar{w})$, letting $\mathcal{N}^G(u_i) = \{v_1, \ldots, v_r\}$ be the enumeration of the neighbourhood of $u_i$ given by the isomorphism, we have that:

  $$\llbracket \pi(\bar{u}) \rrbracket = \mathrm{WMean}_T(\llbracket \rho(\bar{u}, v_1) \rrbracket, \ldots, \llbracket \rho(\bar{u}, v_r) \rrbracket, \llbracket \eta(\bar{u}, v_1) \rrbracket, \ldots, \llbracket \eta(\bar{u}, v_r) \rrbracket) \qquad (\Box)$$

  For any $y_j \in \mathcal{N}(w_i)$, let:

  $$T_j := \mathcal{N}_{\ell-1}^T(\bar{w}, y_j)$$
  $$T_j^\rho := \mathcal{N}_{\mathrm{Reach}(\rho)}^T(\bar{w}, y_j)$$
  $$T_j^\eta := \mathcal{N}_{\mathrm{Reach}(\eta)}^T(\bar{w}, y_j)$$

Further, for any $\bar{a} \in ([0,1]^d)^{|T|}$ let $\bar{a}_j^\rho$ and $\bar{a}_j^\eta$ be the tuple of elements of $\bar{a}$ corresponding to the nodes of $T_j^\rho$ and $T_j^\eta$ respectively.

Let:
$$e_\pi^T(\bar{a}) := \mathrm{WMean}_T\left(e_\rho^{T_1^\rho}(\bar{a}_1^\rho), \ldots, e_\rho^{T_r^\rho}(\bar{a}_r^\rho), e_\eta^{T_1^\eta}(\bar{a}_1^\eta), \ldots, e_\eta^{T_r^\eta}(\bar{a}_r^\eta),\right)$$

Take $\epsilon, \theta > 0$. Applying the induction hypothesis to the term $\rho(\bar{u}, v)$ and to $\eta(\bar{u}, v)$, which have Reach at most $\ell - 1$, and using the fact that $\mathcal{N}^T(w_i)$ is finite, there is $N$ such that for all $n \geq N$ with probability at least $1 - \theta$, for every $y_j \in \mathcal{N}^T(w_i)$ and every $(k+1)$-tuple of nodes $(\bar{u}, v)$ such that $\mathcal{N}_{\ell-1}^G(\bar{u}, v) \cong (T_j, \bar{w}, y_j)$ we have that both:

$$\left\|e_\rho^{T_j^\rho}(H_G(s_1), \ldots, H_G(s_{|T_j^\rho|})) - [\![\rho(\bar{u}, v)]\!]\right\| < \epsilon$$

and:

$$\left\|e_\eta^{T_j^\eta}(H_G(s_1), \ldots, H_G(s_{|T_j^\eta|})) - [\![\eta(\bar{u}, v)]\!]\right\| < \epsilon$$

Under this event, by equation ($\square$) and the Lipschitzness of $\mathrm{WMean}_T$ we have:

$$\left\|e_\pi^T(H_G(s_1), \ldots, H_G(s_{|T|})) - [\![\pi(\bar{u})]\!]\right\| < L_T\epsilon$$

where $L_T$ is the Lipschitz constant of $\mathrm{WMean}_T$.

- **Inductive step for the GCN aggregator** $\pi = \mathrm{GCN}_{v \in \mathcal{N}(u_i)}\rho$.

  This step is very similar to the previous, where we again use the fact that the value of a local aggregator is determined by the values of the terms it is aggregating in the local neighbourhood. The only difference is that we now have a different Lipschitz function:

  $$\mathrm{GCN}_T(\boldsymbol{a}_1, \ldots, \boldsymbol{a}_r) := \sum_{j=1}^r \frac{1}{|\mathcal{N}^T(u_i)||\mathcal{N}^T(y_j)|}\boldsymbol{a}_j$$

  The rest of the proof is as before.

- **Inductive step for global weighted mean** $\pi = \sum_v \rho \star h(\eta)$.

  Note that $\ell = 0$. Let $\ell' = \max(\mathrm{Reach}(\rho), \mathrm{Reach}(\eta))$. Take a $k$-rooted graph $(T, \bar{w})$. For any tuple $\bar{u}$ and $(k+1)$-rooted graph $(T', \bar{w}, y)$ extending $(T, \bar{w})$ let:

  $$[\![T'(\bar{u})]\!] := |\{v \mid \mathcal{N}_{\ell'}^G(\bar{u}, v) \cong (T', \bar{w}, y)\}|$$

  By the parameterized Weak Local Convergence result, Theorem D.4, for every such $T'$ extending there is a relative neighborhood weight $q_{T|T'} \in [0,1]$ such that for all $\epsilon, \theta > 0$ there is $N \in \mathbb{N}$ such that for all $n \geq N$ with probability at least $1 - \theta$, for every $k$-tuple $\bar{u}$ such that $\mathcal{N}_\ell(\bar{u}) = T$ we have that:

  $$\left|\frac{[\![T'(\bar{u})]\!]}{n} - q_{T|T'}\right| < \epsilon$$

  and moreover the relative neighborhood weights sum to one:

  $$\sum_{T' \in \mathsf{Ext}(T)} q_{T|T'} = 1$$

  Consider any $(T', \bar{w}, y) \in \mathsf{Ext}(T)$. In order to define the controller, we need to identify the nodes in the canonical enumeration corresponding to $\mathcal{N}_{\ell'-1}^T(\bar{u})$. Let:

  $$r_T := \left|\mathcal{N}_{\ell'-1}^T(\bar{u})\right|$$

  Given tuples of $\mathbb{R}^d$-vectors $\bar{a} = (\boldsymbol{a}_1, \ldots, \boldsymbol{a}_{r_T})$ and $\bar{b} = (\boldsymbol{b}_{r_T+1}, \ldots, \boldsymbol{b}_{|T'|})$, let $\mathrm{Arrange}_{T'}(\bar{a}, \bar{b})$ be the $|T'|$-tuple of vectors obtained by assigning the $\boldsymbol{a}_i$'s to the nodes in $\mathcal{N}_{\ell'-1}^T(\bar{u})$ and the $\boldsymbol{b}_i$'s to the remaining nodes in $T'$, according to the canonical enumeration of $T'$.

  Define the controller:

  $$e_\pi^T(\bar{a}) := \frac{f_\pi^T(\bar{a})}{g_\pi^T(\bar{a})}$$

where:

$$f_\pi^T(\bar{\boldsymbol{a}}) := \sum_{T' \in \mathsf{Ext}(T)} \mathbb{E}_{\bar{\boldsymbol{b}} \sim \mathbb{F}} \left[ e_\rho^{T'}(\mathrm{Arrange}_{T'}(\bar{\boldsymbol{a}}, \bar{\boldsymbol{b}})) \cdot h(e_\eta^{T'}(\mathrm{Arrange}_{T'}(\bar{\boldsymbol{a}}, \bar{\boldsymbol{b}}))) \right] \cdot q_{T|T'}$$

and:

$$g_\pi^T(\boldsymbol{a}_1, \ldots, \boldsymbol{a}_k) := \sum_{T' \in \mathsf{Ext}(T)} \mathbb{E}_{\bar{\boldsymbol{b}} \sim \mathbb{F}} \left[ h(e_\eta^{T'}(\mathrm{Arrange}_{T'}(\bar{\boldsymbol{a}}, \bar{\boldsymbol{b}}))) \right] \cdot q_{T|T'}$$

where in both cases the expectation is taken over $\boldsymbol{b}_i$'s sampled independently from node feature distribution $\mathbb{F}$.

Since:

$$e_\rho^{T'}(\boldsymbol{a}_1, \ldots, \boldsymbol{a}_{|T'|}) h(e_\eta^{T'}(\boldsymbol{a}_1, \ldots, \boldsymbol{a}_{|T'|}))$$

and:

$$h(e_\eta^{T'}(\boldsymbol{a}_1, \ldots, \boldsymbol{a}_{|T'|}))$$

are bounded, and $\sum_{T' \in \mathsf{Ext}(T)} q_{T|T'}$ converges, these sums converges absolutely, and hence the controller is well-defined.

This completes the definition of the controller. We now present the proof that it is correct.

Take $\epsilon, \theta > 0$. Take a finite $S \subseteq \mathsf{Ext}(T)$ such that:

$$\sum_{T' \in S} q_{T|T'} > 1 - \epsilon \tag{$\bigcirc$}$$

and each $q_{T|T'} > 0$ for $T' \in S$.

The guarantee given when we applied the parameterized version of Weak Local Convergence, Theorem D.4, is that there is $N_1$ such that for all $n \geq N_1$ with probability at least $1 - \theta$, for all $T' \in S$ and every $k$-tuple $\bar{u}$ such that $\mathcal{N}_\ell(\bar{u}) = T$ we have that:

$$\left| \frac{[\![T'(\bar{u})]\!]}{n} - q_{T|T'} \right| < \epsilon \tag{$\ddagger$}$$

Define the function:

$$f_\pi^\circ(\bar{\boldsymbol{a}}, \bar{\boldsymbol{b}}) := e_\rho^{T'}(\mathrm{Arrange}_{T'}(\bar{\boldsymbol{a}}, \bar{\boldsymbol{b}})) \cdot h(e_\eta^{T'}(\mathrm{Arrange}_{T'}(\bar{\boldsymbol{a}}, \bar{\boldsymbol{b}})))$$

and similarly:

$$g_\pi^\circ(\bar{\boldsymbol{a}}, \bar{\boldsymbol{b}}) := h(e_\eta^{T'}(\mathrm{Arrange}_{T'}(\bar{\boldsymbol{a}}, \bar{\boldsymbol{b}})))$$

Note that these are just the integrands used in $f_\pi^T(\bar{\boldsymbol{a}})$ and $g_\pi^T(\bar{\boldsymbol{a}})$.

Now, for any $(k + 1)$-tuple of nodes $(\bar{u}, v)$ let $\bar{\boldsymbol{a}}^{\bar{u}}$ be the tuple of node features of $\mathcal{N}_{\ell'-1}^G(\bar{u})$ and $\bar{\boldsymbol{b}}^{(\bar{u},v)}$ be the tuple of node features of the remaining nodes in $\mathcal{N}_{\ell'}^G(\bar{u}, v)$, ordered as in the canonical enumeration of $\mathcal{N}_{\ell'}^G(\bar{u}, v)$.

Note that $f_\pi^\circ$ is bounded, say by $\Lambda$. Then for every $\gamma > 0$, by Hoeffding's Inequality (Theorem B.4) for any tuple $\bar{u}$ such that $\mathcal{N}_\ell(\bar{u}) = T$ and $T' \in S$ we have that:

$$\mathbb{P}\left( \left\| \sum_{\substack{v \\ \mathcal{N}_{\ell'}^G(\bar{u},v) \cong T'}} f_\pi^\circ(\bar{\boldsymbol{a}}^{\bar{u}}, \bar{\boldsymbol{b}}^{(\bar{u},v)}) - [\![T'(\bar{u})]\!] \cdot \mathbb{E}_{\bar{\boldsymbol{b}} \sim \mathbb{F}} \left[ f_\pi^\circ(\bar{\boldsymbol{a}}^{\bar{u}}, \bar{\boldsymbol{b}}) \right] \right\| \geq [\![T'(\bar{u})]\!] \gamma \right)$$

$$\leq 2 \exp\left( -\frac{2[\![T'(\bar{u})]\!] \gamma^2}{\Lambda^2} \right)$$

$$\leq 2 \exp\left( -\frac{2n(q'_T - \epsilon) \gamma^2}{\Lambda^2} \right)$$

where in the last inequality we use equation (‡). Hence, taking a union bound, there is $N_2$ such that for all $n \geq N_2$ with probability at least $1 - \theta$, for all $\bar{u}$ such that $\mathcal{N}_\ell(\bar{u}) = T$ and $T' \in \mathsf{Ext}(T)$ we have:

$$\left\| \sum_{\substack{v \\ \mathcal{N}^G_{\ell'}(\bar{u},v) \cong T'}} f^\circ_\pi(\bar{\boldsymbol{a}}^{\bar{u}}, \bar{\boldsymbol{b}}^{(\bar{u},v)}) - |[\![T'(\bar{u})]\!]| \cdot \mathop{\mathbb{E}}_{\bar{\boldsymbol{b}} \sim \mathbb{F}} \left[ f^\circ_\pi(\bar{\boldsymbol{a}}^{\bar{u}}, \bar{\boldsymbol{b}}) \right] \right\| < |[\![T'(\bar{u})]\!]| \gamma$$

If we divide both sides of the inequality through by $n$, take a $\gamma$ below 1, and apply equation (‡) again, we conclude that there is $N_4$ such that for all $n \geq N_4$ with probability at least $1 - \theta$, for all $\bar{u}$ such that $\mathcal{N}_\ell(\bar{u}) = T$ and $T' \in \mathsf{Ext}(T)$ we have:

$$\left\| \frac{1}{n} \sum_{\substack{v \\ \mathcal{N}^G_{\ell'}(\bar{u},v) \cong T'}} f^\circ_\pi(\bar{\boldsymbol{a}}^{\bar{u}}, \bar{\boldsymbol{b}}^{(\bar{u},v)}) - q_{T|T'} \mathop{\mathbb{E}}_{\bar{\boldsymbol{b}} \sim \mathbb{F}} \left[ f^\circ_\pi(\bar{\boldsymbol{a}}^{\bar{u}}, \bar{\boldsymbol{b}}) \right] \right\| < \epsilon \qquad (\triangle_f)$$

Similarly, there is $N_5$ such that for all $n \geq N_5$ with probability at least $1 - \theta$, for all $\bar{u}$ such that $\mathcal{N}_\ell(\bar{u}) = T$ and $T' \in \mathsf{Ext}(T)$ we have:

$$\left\| \frac{1}{n} \sum_{\substack{v \\ \mathcal{N}^G_{\ell'}(\bar{u},v) \cong T'}} g^\circ_\pi(\bar{\boldsymbol{a}}^{\bar{u}}, \bar{\boldsymbol{b}}^{(\bar{u},v)}) - q_{T|T'} \mathop{\mathbb{E}}_{\bar{\boldsymbol{b}} \sim \mathbb{F}} \left[ g^\circ_\pi(\bar{\boldsymbol{a}}^{\bar{u}}, \bar{\boldsymbol{b}}) \right] \right\| < \epsilon \qquad (\triangle_g)$$

By the induction hypothesis, there is $N_6$ such that for all $n \geq N_6$ with probability at least $1 - \theta$, for every $T' \in S$ and every $(k+1)$-tuple of nodes $(\bar{u}, v)$ such that $\mathcal{N}^G_{\ell'}(\bar{u}, v) \cong T'$ we have that both:

$$\left\| e^{T'}_\rho(H_G(s_1), \ldots, H_G(s_{|T'|})) - [\![\rho(\bar{u}, v)]\!] \right\| < \epsilon \qquad (\dagger_\rho)$$

and:

$$\left\| e^{T'}_\eta(H_G(s_1), \ldots, H_G(s_{|T'|})) - [\![\eta(\bar{u}, v)]\!] \right\| < \epsilon \qquad (\dagger_\eta)$$

Let $N := \max(N_1, N_2, N_3, N_4, N_5, N_6)$ and take $n \geq N$. Then for such $n$ the probability that these six events happen is at least $1 - 6\theta$. We will condition on this.

Take any $k$-tuple of nodes $\bar{u}$ such that $\mathcal{N}_\ell(\bar{u}) = T$. It suffices to show, using the definition of the interpretation of the weighted mean operator, that:

$$\left\| \frac{\sum_v [\![\rho(\bar{u}, v)]\!] h([\![\eta(\bar{u}, v)]\!])}{\sum_v h([\![\eta(\bar{u}, v)]\!])} - \frac{f^T_\pi(H_G(u_1), \ldots, H_G(u_k))}{g^T_\pi(H_G(u_1), \ldots, H_G(u_k))} \right\| < \iota$$

for some $\iota$ which we can make arbitrarily small by choosing $\epsilon$ small enough.

As above it suffices to find $\zeta, \xi > 0$ which we can make arbitrarily small and constants $\nu, \Omega > 0$ such that:

$$\left\| \frac{1}{n} \sum_v [\![\rho(\bar{u}, v)]\!] h([\![\eta(\bar{u}, v)]\!]) - f^T_\pi(H_G(s_1), \ldots, H_G(s_{|T|})) \right\| < \zeta \qquad (6)$$

$$\left\| \frac{1}{n} \sum_v h([\![\eta(\bar{u}, v)]\!]) - g^T_\pi(H_G(s_1), \ldots, H_G(s_{|T|})) \right\| < \xi \qquad (7)$$

$$\left\| \sum_v [\![\rho(\bar{u}, v)]\!] h([\![\eta(\bar{u}, v)]\!]) \right\| < \Omega n \qquad (8)$$

$$\left\| \sum_v h([\![\eta(\bar{u}, v)]\!]) \right\| < \Omega n \qquad (9)$$

$$\forall i \leq d \colon \left[ \sum_v h([\![\eta(\bar{u}, v)]\!]) \right]_i > \nu n \qquad (10)$$

Equation (8), equation (9) and equation (10) follow as before. We show equation (7), the proof of equation (6) being similar.

By equation $(\dagger_\eta)$ we have that for every $v$:

$$\left\| h([\![\eta(\bar{u}, v)]\!]) - g_\pi^\circ(\bar{\boldsymbol{a}}^{\bar{u}}, \bar{\boldsymbol{b}}^{(\bar{u},v)}) \right\| < \epsilon$$

Hence:

$$\left\| \frac{1}{n} \sum_v h([\![\eta(\bar{u}, v)]\!]) - \frac{1}{n} \sum_v g_\pi^\circ(\bar{\boldsymbol{a}}^{\bar{u}}, \bar{\boldsymbol{b}}^{(\bar{u},v)}) \right\| < \epsilon$$

Now:

$$\frac{1}{n} \sum_v g_\pi^\circ(\bar{\boldsymbol{a}}^{\bar{u}}, \bar{\boldsymbol{b}}^{(\bar{u},v)}) = \frac{1}{n} \sum_{T' \in \mathsf{Ext}(T)} \sum_{\substack{v \\ \mathcal{N}_{\ell'}^G(\bar{u},v) \cong T'}} g_\pi^\circ(\bar{\boldsymbol{a}}^{\bar{u}}, \bar{\boldsymbol{b}}^{(\bar{u},v)})$$

Letting $\Lambda$ be a bound on the norm of $g_\pi^\circ(\bar{\boldsymbol{a}}^{\bar{u}}, \bar{\boldsymbol{b}}^{(\bar{u},v)})$, by equation $(\bigcirc)$ we have that:

$$\left\| \frac{1}{n} \sum_v g_\pi^\circ(\bar{\boldsymbol{a}}^{\bar{u}}, \bar{\boldsymbol{b}}^{(\bar{u},v)}) - \frac{1}{n} \sum_{T' \in S} \sum_{\substack{v \\ \mathcal{N}_{\ell'}^G(\bar{u},v) \cong T'}} g_\pi^\circ(\bar{\boldsymbol{a}}^{\bar{u}}, \bar{\boldsymbol{b}}^{(\bar{u},v)}) \right\| < \epsilon\Lambda$$

By equation $(\triangle_g)$ we have that:

$$\left\| \frac{1}{n} \sum_{T' \in S} \sum_{\substack{v \\ \mathcal{N}_{\ell'}^G(\bar{u},v) \cong T'}} g_\pi^\circ(\bar{\boldsymbol{a}}^{\bar{u}}, \bar{\boldsymbol{b}}^{(\bar{u},v)}) - \frac{1}{n} \sum_{T' \in S} q_{T|T'} \mathop{\mathbb{E}}_{\bar{\boldsymbol{b}} \sim \mathbb{F}} \left[ g_\pi^\circ(\bar{\boldsymbol{a}}^{\bar{u}}, \bar{\boldsymbol{b}}) \right] \right\| \le \epsilon$$

Finally, by equation $(\bigcirc)$ again we have that:

$$\left\| \frac{1}{n} \sum_{T' \in S} q_{T|T'} \mathop{\mathbb{E}}_{\bar{\boldsymbol{b}} \sim \mathbb{F}} \left[ g_\pi^\circ(\bar{\boldsymbol{a}}^{\bar{u}}, \bar{\boldsymbol{b}}) \right] - g_\pi^T(\bar{\boldsymbol{a}}^{\bar{u}}) \right\| < \epsilon\Lambda$$

This completes the proof of the inductive construction of the controllers, thus proving Theorem 5.6. $\qquad\square$

**Application to prove the final theorem, Theorem 5.1 for the sparse Erdős-Rényi and Barabási-Albert cases.** To complete the proof, we note that the term $\tau$ has no free variables, and as a subterm of itself has reach 0. The controller $e_\tau^\varnothing$ of is a constant $\boldsymbol{z}$, and hence by the induction hypothesis applied to it, for every $\epsilon, \theta > 0$ there is $N \in \mathbb{N}$ such that for all $n \ge N$ with probability at least $1 - \theta$ we have that:

$$\| [\![\tau]\!] - \boldsymbol{z} \| < \epsilon$$

# E   Proof for the stochastic block model

We now prove the convergence result for the stochastic block model. The proof follows the same structure as the density case in Theorem 5.1, except that the notion of graph type is augmented slightly. We therefore only indicate the differences in the proof.

For this is it helpful to be able to remember the community to which each node belongs. Given any graph $G$ generated by the stochastic block model and node $v$, let $C(v) \in \{1, \ldots, m\}$ be the community to which $v$ belongs.

To prove the result, we first need that the random work positional encodings converge, as in Lemma C.8. In fact they converge to $\boldsymbol{0}$.

**Lemma E.1.** *Let $n_1, \ldots, n_m \colon \mathbb{N} \to \mathbb{N}$ and $\boldsymbol{P}$ be as in Theorem 5.1. Then for every $k \in \mathbb{N}$ and $\epsilon, \theta > 0$ there is $N \in \mathbb{N}$ such that for all $n \ge N$ with probability at least $1 - \theta$, for all nodes $v$ we have that:*

$$\| rw_k(v) \| < \epsilon$$

*Proof.* For each $j \in \{1, \ldots, m\}$, let $\frac{n_j}{n}$ converge to $q_j$. Let:

$$r_j := q_1 \boldsymbol{P}_{1,j} + \cdots + q_m \boldsymbol{P}_{m,j}$$

Note that $nr_j$ is the expected degree of a node in community $j$. By Hoeffding's Inequality (Theorem B.4) and a union bound, there is $N$ such that for all $n \geq N$ with probability at least $1 - \theta$, for all $v$ we have that:

$$\left| \frac{d(v)}{n} - r_{C(v)} \right| < \epsilon$$

Take $n \geq N$ and condition on this event. Take any node $v$. When $r_{C(v)} = 0$, the node $v$ has no neighbours, and so $\mathrm{rw}_k(v) = \boldsymbol{0}$.

So assume $r_{C(v)} > 0$. Let $j = C(v)$. Then as in the proof of Lemma C.8 we can show that the proportion of random walks of length $k$ starting at $v$ is at most:

$$\frac{(\tilde{p} + \epsilon)^{k-1}}{(\tilde{p} - \epsilon)^k} n^{-1}$$

which converges to 0 as $n \to \infty$. $\qquad\square$

We now follow the structure of the proof for the Erdős-Rényi dense cases. This time we augment our vocabulary for atomic types with a predicate $P_j$ for each community $j$, so that $P_j(v)$ holds if and only if $v$ belongs to community $j$.

With this we define the community atomic type of a tuple of nodes $\bar{u}$ in graph, notation $\mathrm{ComTp}(\bar{u})$ to be the atomic formulas satisfied by $\bar{u}$ in the language augmented with the $P_j$ predicates. For $k \in \mathbb{N}$ let $\mathsf{ComTp}_k$ be the set of all complete community atomic types with $k$ free variables.

For each $j \in \{1, \ldots, m\}$, let $\frac{n_j}{n}$ converge to $q_j$. For any type $t(\bar{u})$ and $t'(\bar{u}, v) \in \mathsf{Ext}(t)$, let:

$$\alpha(t, t') := q_{C(v)}$$

Further, for any type $t(\bar{u})$, free variable $u_i$ in $t$ and $t'(\bar{u}, v) \in \mathsf{Ext}_{u_i}(t)$, let:

$$\alpha_{u_i}(t, t') := \boldsymbol{P}_{C(u_i), C(v)} q_{C(v)}$$

For any term $\pi$ with $k$ free variables, the *feature-type controller*:

$$e_\pi \colon ([0,1]^d)^k \times \mathsf{ComTp}_k \to [0,1]^d$$

is defined exactly as as in the proof of the density case of Theorem 5.1, using the extension proportions $\alpha(t, t')$ and $\alpha_{u_i}(t, t')$ just defined.

We show by induction that every $\epsilon, \delta, \theta > 0$ and $\wedge$-desc $\psi(\bar{u})$, there is $N \in \mathbb{N}$ such that for all $n \geq N$, with probability at least $1 - \theta$ in the space of graphs of size $n$, out of all the tuples $\bar{u}$ such that $\psi(\bar{u})$, at least $1 - \delta$ satisfy:

$$\| e_\pi(H_G(\bar{u}), \mathrm{GrTp}(\bar{u})) - [\![\pi(\bar{u})]\!] \| < \epsilon$$

We then proceed as in the proof of the non-sparse Erdős-Rényi cases of Theorem 5.1. The only difference is that when showing equation $(\triangle_f)$ and equation $(\triangle_g)$ we use the fact that the expected proportion of type extensions $t'(\bar{u}, v)$ of a type $t(\bar{u})$ at a node $u_j$ is $\alpha(t, t')$.

## F  Additional experiments

In this section we provide additional experiments using MeanGNN, GCN [30], GAT [49], and GPS+RW with random walks of length up to 5, over the distributions $\mathrm{ER}(n, p(n) = 0.1)$, $\mathrm{ER}(n, p(n) = \frac{\log n}{n})$, $\mathrm{ER}(n, p(n) = \frac{1}{50n})$ and $\mathrm{BA}(n, m = 5)$. We also experiment with an SBM of 10 communities with equal size, where an edge between nodes within the same community is included with probability $0.7$ and an edge between nodes of different communities is included with probability $0.1$.

**Setup.** Our setup is carefully designed to eliminate confounding factors:

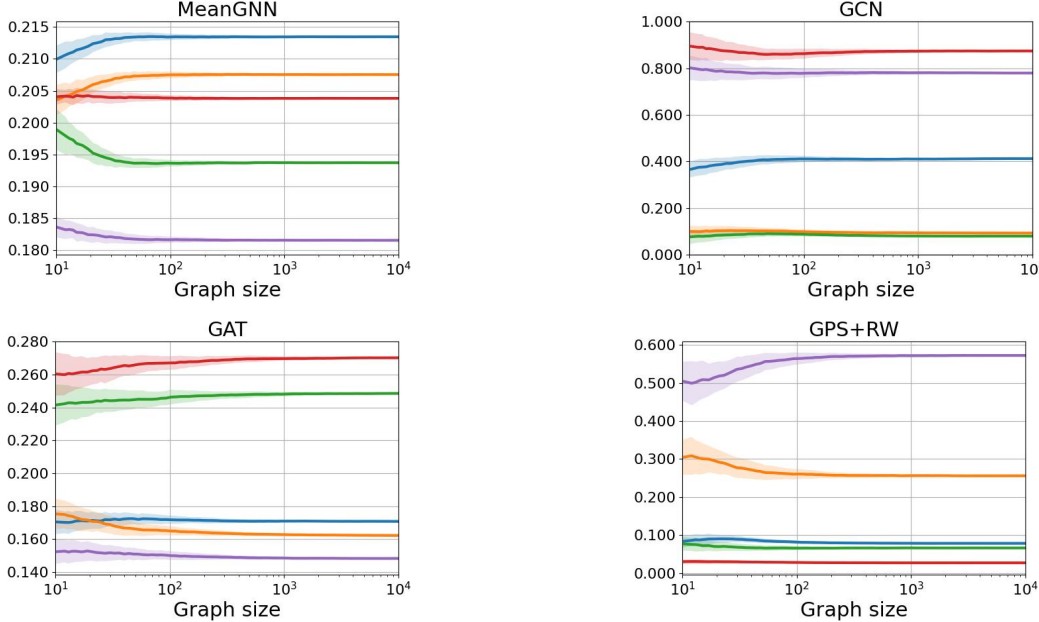

Figure 6: The 5 class probabilities (in different colours) of a MEANGNN, GCN, GAT and GPS+RW model initialization over the $\mathrm{ER}(n, p(n) = 0.1)$ graph distributions, as we draw increasingly larger graphs.

- We consider 5 models with the same architecture, each having randomly initialized weights, utilizing a ReLU non-linearity, and applying a softmax function to their outputs. Each model uses a hidden dimension of 128, 3 layers and an output dimension of 5.
- We draw graphs of sizes up to 10,000, where we take 100 samples of each graph size. Node features are independently drawn from $U[0, 1]$ and the initial feature dimension is 128.

Further details are available in the experiments repository at `https://github.com/benfinkelshtein/GNN-Asymptotically-Constant`.

Much like in Section 6, the convergence of class probabilities is apparent across all models and graph distributions, in accordance with our main theorems (**Q1**). We again observe that attention-based models such as GAT and GPS+RW exhibit delayed convergence and greater standard deviation in comparison to MeanGNN and GCN, which further strengthening our previous conclusions.

## G   Acknowledgements

This research was funded in whole or in part by EPSRC grant EP/T022124/1. For the purpose of Open Access, the authors have applied a CC BY public copyright license to any Author Accepted Manuscript (AAM) version arising from this submission.


Figure 7: The 5 class probabilities (in different colours) of a MEANGNN, GCN, GAT and GPS+RW model initialization over the $\mathrm{ER}(n, p(n) = \frac{\log n}{n})$ graph distributions, as we draw increasingly larger graphs.

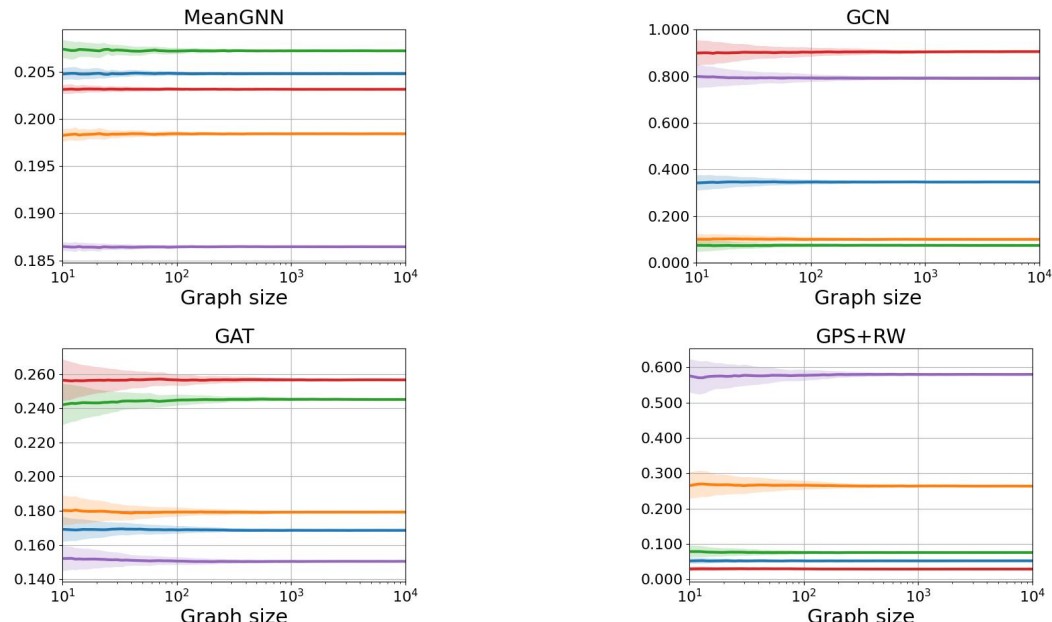

Figure 8: The 5 class probabilities (in different colours) of a MEANGNN, GCN, GAT and GPS+RW model initialization over the $\mathrm{ER}(n, p(n) = \frac{1}{50n})$ graph distributions, as we draw increasingly larger graphs.

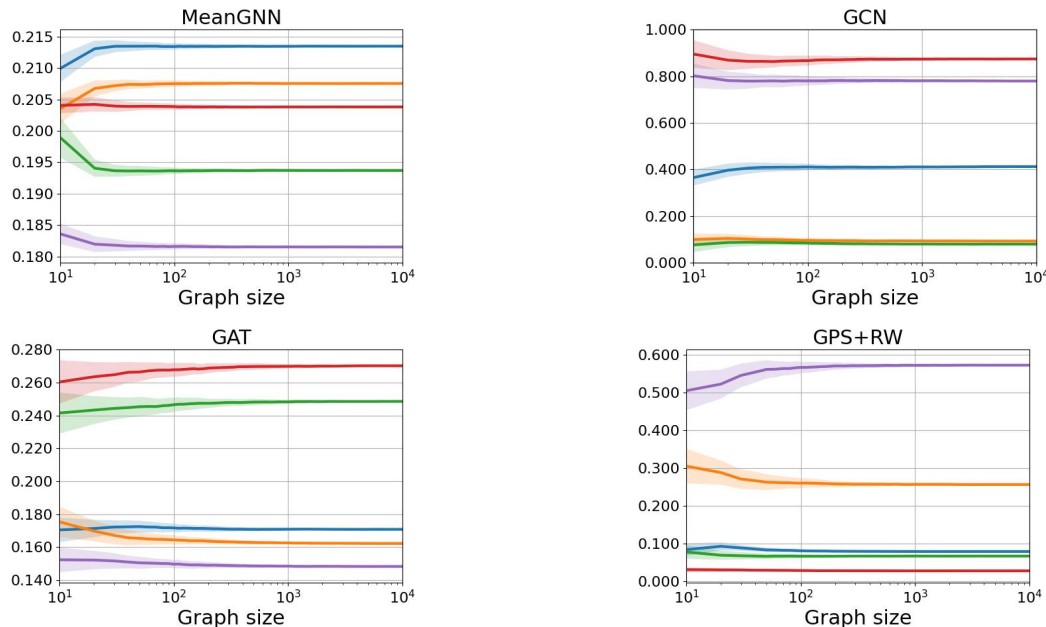

Figure 9: The 5 class probabilities (in different colours) of a MEANGNN, GCN, GAT and GPS+RW model initialization over the SBM graph distributions, as we draw increasingly larger graphs.

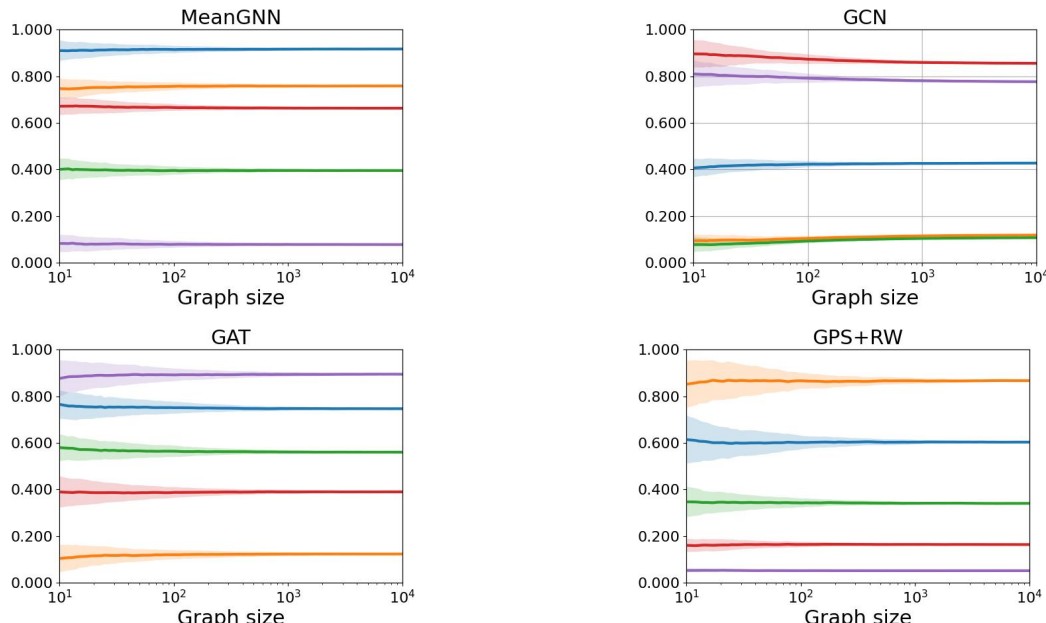

Figure 10: The 5 class probabilities (in different colours) of a MEANGNN, GCN, GAT and GPS+RW model initialization over the $\mathrm{BA}(n, m = 5)$ graph distributions, as we draw increasingly larger graphs.

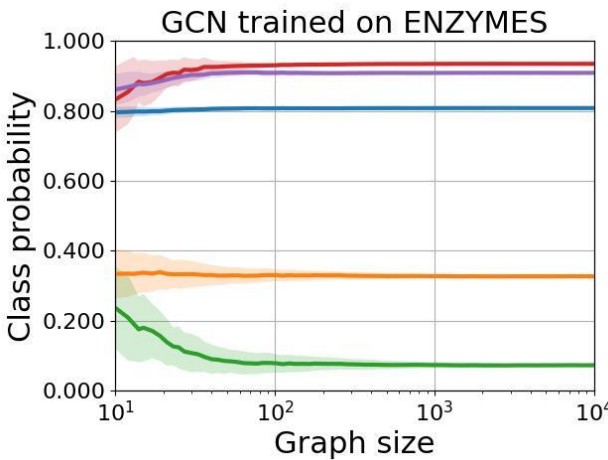

Figure 11: A three-layer GCN with hidden dimension 128 is trained on the ENZYMES dataset with one class removed so that there are five output classes. This model is then run on graphs drawn from $\text{ER}(n, p(n) = 0.1)$ for increasing sizes $n$, and the mean output probabilities are recorded, along with standard deviation

- The answer NA means that the abstract and introduction do not include the claims made in the paper.
- The abstract and/or introduction should clearly state the claims made, including the contributions made in the paper and important assumptions and limitations. A No or NA answer to this question will not be perceived well by the reviewers.
- The claims made should match theoretical and experimental results, and reflect how much the results can be expected to generalize to other settings.
- It is fine to include aspirational goals as motivation as long as it is clear that these goals are not attained by the paper.

2. **Limitations**

Question: Does the paper discuss the limitations of the work performed by the authors?

Answer: [Yes]

Justification: We discuss limitations in the discussion section: in particular we note that a.a.s. convergence does not apply universally to every GNN, and it does not apply for arbitrary instantiations of popular random graph models (like Erdos Renyi), regardless of the controlling parameters.

Guidelines:

- The answer NA means that the paper has no limitation while the answer No means that the paper has limitations, but those are not discussed in the paper.
- The authors are encouraged to create a separate "Limitations" section in their paper.
- The paper should point out any strong assumptions and how robust the results are to violations of these assumptions (e.g., independence assumptions, noiseless settings, model well-specification, asymptotic approximations only holding locally). The authors should reflect on how these assumptions might be violated in practice and what the implications would be.
- The authors should reflect on the scope of the claims made, e.g., if the approach was only tested on a few datasets or with a few runs. In general, empirical results often depend on implicit assumptions, which should be articulated.
- The authors should reflect on the factors that influence the performance of the approach. For example, a facial recognition algorithm may perform poorly when image resolution is low or images are taken in low lighting. Or a speech-to-text system might not be used reliably to provide closed captions for online lectures because it fails to handle technical jargon.
- The authors should discuss the computational efficiency of the proposed algorithms and how they scale with dataset size.

- If applicable, the authors should discuss possible limitations of their approach to address problems of privacy and fairness.
- While the authors might fear that complete honesty about limitations might be used by reviewers as grounds for rejection, a worse outcome might be that reviewers discover limitations that aren't acknowledged in the paper. The authors should use their best judgment and recognize that individual actions in favor of transparency play an important role in developing norms that preserve the integrity of the community. Reviewers will be specifically instructed to not penalize honesty concerning limitations.

3. **Theory Assumptions and Proofs**

   Question: For each theoretical result, does the paper provide the full set of assumptions and a complete (and correct) proof?

   Answer: [Yes]

   Justification: full proofs are provided in the appendix to the paper.

   Guidelines:
   - The answer NA means that the paper does not include theoretical results.
   - All the theorems, formulas, and proofs in the paper should be numbered and cross-referenced.
   - All assumptions should be clearly stated or referenced in the statement of any theorems.
   - The proofs can either appear in the main paper or the supplemental material, but if they appear in the supplemental material, the authors are encouraged to provide a short proof sketch to provide intuition.
   - Inversely, any informal proof provided in the core of the paper should be complemented by formal proofs provided in appendix or supplemental material.
   - Theorems and Lemmas that the proof relies upon should be properly referenced.

4. **Experimental Result Reproducibility**

   Question: Does the paper fully disclose all the information needed to reproduce the main experimental results of the paper to the extent that it affects the main claims and/or conclusions of the paper (regardless of whether the code and data are provided or not)?

   Answer: [Yes]

   Justification: we detail the experimental settings and datasets in the paper, and provide a link to a github with information about how to reproduce the experiments.

   Guidelines:
   - The answer NA means that the paper does not include experiments.
   - If the paper includes experiments, a No answer to this question will not be perceived well by the reviewers: Making the paper reproducible is important, regardless of whether the code and data are provided or not.
   - If the contribution is a dataset and/or model, the authors should describe the steps taken to make their results reproducible or verifiable.
   - Depending on the contribution, reproducibility can be accomplished in various ways. For example, if the contribution is a novel architecture, describing the architecture fully might suffice, or if the contribution is a specific model and empirical evaluation, it may be necessary to either make it possible for others to replicate the model with the same dataset, or provide access to the model. In general. releasing code and data is often one good way to accomplish this, but reproducibility can also be provided via detailed instructions for how to replicate the results, access to a hosted model (e.g., in the case of a large language model), releasing of a model checkpoint, or other means that are appropriate to the research performed.
   - While NeurIPS does not require releasing code, the conference does require all submissions to provide some reasonable avenue for reproducibility, which may depend on the nature of the contribution. For example
     (a) If the contribution is primarily a new algorithm, the paper should make it clear how to reproduce that algorithm.
     (b) If the contribution is primarily a new model architecture, the paper should describe the architecture clearly and fully.
     (c) If the contribution is a new model (e.g., a large language model), then there should either be a way to access this model for reproducing the results or a way to reproduce the model (e.g., with an open-source dataset or instructions for how to construct the dataset).

(d) We recognize that reproducibility may be tricky in some cases, in which case authors are welcome to describe the particular way they provide for reproducibility. In the case of closed-source models, it may be that access to the model is limited in some way (e.g., to registered users), but it should be possible for other researchers to have some path to reproducing or verifying the results.

5. **Open access to data and code**

Question: Does the paper provide open access to the data and code, with sufficient instructions to faithfully reproduce the main experimental results, as described in supplemental material?

Answer: [Yes]

Justification: all of the necessary information is available in the GitHub repository.

Guidelines:

- The answer NA means that paper does not include experiments requiring code.
- Please see the NeurIPS code and data submission guidelines (`https://nips.cc/public/guides/CodeSubmissionPolicy`) for more details.
- While we encourage the release of code and data, we understand that this might not be possible, so "No" is an acceptable answer. Papers cannot be rejected simply for not including code, unless this is central to the contribution (e.g., for a new open-source benchmark).
- The instructions should contain the exact command and environment needed to run to reproduce the results. See the NeurIPS code and data submission guidelines (`https://nips.cc/public/guides/CodeSubmissionPolicy`) for more details.
- The authors should provide instructions on data access and preparation, including how to access the raw data, preprocessed data, intermediate data, and generated data, etc.
- The authors should provide scripts to reproduce all experimental results for the new proposed method and baselines. If only a subset of experiments are reproducible, they should state which ones are omitted from the script and why.
- At submission time, to preserve anonymity, the authors should release anonymized versions (if applicable).
- Providing as much information as possible in supplemental material (appended to the paper) is recommended, but including URLs to data and code is permitted.

6. **Experimental Setting/Details**

Question: Does the paper specify all the training and test details (e.g., data splits, hyperparameters, how they were chosen, type of optimizer, etc.) necessary to understand the results?

Answer: [Yes]

Justification: we provide all the necessary information on the datasets and the models used (our submission does not deal with training).

Guidelines:

- The answer NA means that the paper does not include experiments.
- The experimental setting should be presented in the core of the paper to a level of detail that is necessary to appreciate the results and make sense of them.
- The full details can be provided either with the code, in appendix, or as supplemental material.

7. **Experiment Statistical Significance**

Question: Does the paper report error bars suitably and correctly defined or other appropriate information about the statistical significance of the experiments?

Answer: [Yes]

Justification: we provide envelope error regions for Figure 3. For the figures plotting standard deviations across graph sizes, we judged that it would be clearer to simply plot all datapoints, rather than take the mean (otherwise we would need to plot the standard deviation of standard deviations).

Guidelines:

- The answer NA means that the paper does not include experiments.
- The authors should answer "Yes" if the results are accompanied by error bars, confidence intervals, or statistical significance tests, at least for the experiments that support the main claims of the paper.

- The factors of variability that the error bars are capturing should be clearly stated (for example, train/test split, initialization, random drawing of some parameter, or overall run with given experimental conditions).
- The method for calculating the error bars should be explained (closed form formula, call to a library function, bootstrap, etc.)
- The assumptions made should be given (e.g., Normally distributed errors).
- It should be clear whether the error bar is the standard deviation or the standard error of the mean.
- It is OK to report 1-sigma error bars, but one should state it. The authors should preferably report a 2-sigma error bar than state that they have a 96% CI, if the hypothesis of Normality of errors is not verified.
- For asymmetric distributions, the authors should be careful not to show in tables or figures symmetric error bars that would yield results that are out of range (e.g. negative error rates).
- If error bars are reported in tables or plots, The authors should explain in the text how they were calculated and reference the corresponding figures or tables in the text.

8. **Experiments Compute Resources**

Question: For each experiment, does the paper provide sufficient information on the computer resources (type of compute workers, memory, time of execution) needed to reproduce the experiments?

Answer: [Yes]

Justification: we provide information on the resources we used, and for replication these could be utilized.

Guidelines:

- The answer NA means that the paper does not include experiments.
- The paper should indicate the type of compute workers CPU or GPU, internal cluster, or cloud provider, including relevant memory and storage.
- The paper should provide the amount of compute required for each of the individual experimental runs as well as estimate the total compute.
- The paper should disclose whether the full research project required more compute than the experiments reported in the paper (e.g., preliminary or failed experiments that didn't make it into the paper).

9. **Code Of Ethics**

Question: Does the research conducted in the paper conform, in every respect, with the NeurIPS Code of Ethics https://neurips.cc/public/EthicsGuidelines?

Answer: [Yes]

Justification: we have reviewed the ethics code and are confident that our paper conforms to it.

Guidelines:

- The answer NA means that the authors have not reviewed the NeurIPS Code of Ethics.
- If the authors answer No, they should explain the special circumstances that require a deviation from the Code of Ethics.
- The authors should make sure to preserve anonymity (e.g., if there is a special consideration due to laws or regulations in their jurisdiction).

10. **Broader Impacts**

Question: Does the paper discuss both potential positive societal impacts and negative societal impacts of the work performed?

Answer: [NA] .

Justification: this is a theoretically-oriented paper on convergence properties of graph neural networks. There is no direct path to negative applications.

Guidelines:

- The answer NA means that there is no societal impact of the work performed.
- If the authors answer NA or No, they should explain why their work has no societal impact or why the paper does not address societal impact.

- Examples of negative societal impacts include potential malicious or unintended uses (e.g., disinformation, generating fake profiles, surveillance), fairness considerations (e.g., deployment of technologies that could make decisions that unfairly impact specific groups), privacy considerations, and security considerations.
- The conference expects that many papers will be foundational research and not tied to particular applications, let alone deployments. However, if there is a direct path to any negative applications, the authors should point it out. For example, it is legitimate to point out that an improvement in the quality of generative models could be used to generate deepfakes for disinformation. On the other hand, it is not needed to point out that a generic algorithm for optimizing neural networks could enable people to train models that generate Deepfakes faster.
- The authors should consider possible harms that could arise when the technology is being used as intended and functioning correctly, harms that could arise when the technology is being used as intended but gives incorrect results, and harms following from (intentional or unintentional) misuse of the technology.
- If there are negative societal impacts, the authors could also discuss possible mitigation strategies (e.g., gated release of models, providing defenses in addition to attacks, mechanisms for monitoring misuse, mechanisms to monitor how a system learns from feedback over time, improving the efficiency and accessibility of ML).

11. **Safeguards**

Question: Does the paper describe safeguards that have been put in place for responsible release of data or models that have a high risk for misuse (e.g., pretrained language models, image generators, or scraped datasets)?

Answer: [NA] .

Justification: we do not provide new datasets or models in this work.

Guidelines:

- The answer NA means that the paper poses no such risks.
- Released models that have a high risk for misuse or dual-use should be released with necessary safeguards to allow for controlled use of the model, for example by requiring that users adhere to usage guidelines or restrictions to access the model or implementing safety filters.
- Datasets that have been scraped from the Internet could pose safety risks. The authors should describe how they avoided releasing unsafe images.
- We recognize that providing effective safeguards is challenging, and many papers do not require this, but we encourage authors to take this into account and make a best faith effort.

12. **Licenses for existing assets**

Question: Are the creators or original owners of assets (e.g., code, data, models), used in the paper, properly credited and are the license and terms of use explicitly mentioned and properly respected?

Answer: [Yes]

Justification: we cite the public datasets which we use.

Guidelines:

- The answer NA means that the paper does not use existing assets.
- The authors should cite the original paper that produced the code package or dataset.
- The authors should state which version of the asset is used and, if possible, include a URL.
- The name of the license (e.g., CC-BY 4.0) should be included for each asset.
- For scraped data from a particular source (e.g., website), the copyright and terms of service of that source should be provided.
- If assets are released, the license, copyright information, and terms of use in the package should be provided. For popular datasets, `paperswithcode.com/datasets` has curated licenses for some datasets. Their licensing guide can help determine the license of a dataset.
- For existing datasets that are re-packaged, both the original license and the license of the derived asset (if it has changed) should be provided.
- If this information is not available online, the authors are encouraged to reach out to the asset's creators.

13. **New Assets**

    Question: Are new assets introduced in the paper well documented and is the documentation provided alongside the assets?

    Answer: [NA]

    Justification: this theoretically-oriented paper does not provide new assets.

    Guidelines:

    - The answer NA means that the paper does not release new assets.
    - Researchers should communicate the details of the dataset/code/model as part of their submissions via structured templates. This includes details about training, license, limitations, etc.
    - The paper should discuss whether and how consent was obtained from people whose asset is used.
    - At submission time, remember to anonymize your assets (if applicable). You can either create an anonymized URL or include an anonymized zip file.

14. **Crowdsourcing and Research with Human Subjects**

    Question: For crowdsourcing experiments and research with human subjects, does the paper include the full text of instructions given to participants and screenshots, if applicable, as well as details about compensation (if any)?

    Answer: [NA] .

    Justification: there are no crowdsourcing experiments in our submission.

    Guidelines:

    - The answer NA means that the paper does not involve crowdsourcing nor research with human subjects.
    - Including this information in the supplemental material is fine, but if the main contribution of the paper involves human subjects, then as much detail as possible should be included in the main paper.
    - According to the NeurIPS Code of Ethics, workers involved in data collection, curation, or other labor should be paid at least the minimum wage in the country of the data collector.

15. **Institutional Review Board (IRB) Approvals or Equivalent for Research with Human Subjects**

    Question: Does the paper describe potential risks incurred by study participants, whether such risks were disclosed to the subjects, and whether Institutional Review Board (IRB) approvals (or an equivalent approval/review based on the requirements of your country or institution) were obtained?

    Answer: [NA] .

    Justification: no human subjects were used.

    Guidelines:

    - The answer NA means that the paper does not involve crowdsourcing nor research with human subjects.
    - Depending on the country in which research is conducted, IRB approval (or equivalent) may be required for any human subjects research. If you obtained IRB approval, you should clearly state this in the paper.
    - We recognize that the procedures for this may vary significantly between institutions and locations, and we expect authors to adhere to the NeurIPS Code of Ethics and the guidelines for their institution.
    - For initial submissions, do not include any information that would break anonymity (if applicable), such as the institution conducting the review.

