# OpenReview forum: "Almost Surely Asymptotically Constant Graph Neural Networks"
_NeurIPS.cc/2024/Conference — NeurIPS 2024 poster_

### Official Review · Reviewer_xvBN · 2024-06-21

**Soundness:** 3
**Presentation:** 3
**Contribution:** 3
**Rating:** 6
**Confidence:** 4

**Summary:**

This paper theoretically analyze the phenomenon that GNN-based probabilistic classifiers almost surely converge to a constant. A formal approach based on term language is proposed. It provides a unified framework to study different GNN models.

**Strengths:**

1. The paper is well-organized.
2. The theoretical study is solid.
3. The main finding is interesting.

**Weaknesses:**

1. The use of ``term language'' is abstract and can be difficult to understand. It is preferable the authors can explain more and provide simple (numerical) examples, e.g, on an explicit small graph.
2. The paper discusses an interesting asymptotic behavior of GNNs as classifiers. However, its impact on GNN real applications is not fully discussed.
3. I do not find a thorough discussion if features are not i.i.d.

**Questions:**

1. Can the authors give an explicit (rigorous) definition of "probabilistic classifier"?
2. In the discussion of almost sure convergence (3.1), what is $\bar{\mu}$?
3. From the definition of ``term language'' (Definition 4.1), it seems that it is a high-level summary of components in GNN architectures. I do not see what the essential properties are that lead to key results such as Theorem 5.1. Can authors elaborate on this?
4. For Theorem 5.1, is the almost sure limit a single (deterministic) vector?
5. I interpret the main findings of the paper as: large graphs (from one of the discussed models) are not distinguishable. Is this correct? Is there any way that the GNN architecture can be modified to overcome such a limitation?
6. Node features are assumed to be i.i.d. (3.1). In applications, this is usually not the case. How will non i.i.d. features affect the results?

**Limitations:**

N.A.

---

> ### Author Rebuttal · Authors · 2024-08-06
>
> Thank you very much for your review and for finding our work solid, well-organised, and interesting. We reply to each of your comments below.
>
> > The use of ``term language'' is abstract and can be difficult to understand. It is preferable the authors can explain more and provide simple (numerical) examples, e.g, on an explicit small graph.
>
> Thank you for this suggestion, which we like very much. In our `supplementary-page.pdf` we have included a graphical example of computing a term on a small graph, which we will include in the camera-ready paper (Figure 3).
>
> > Can the authors give an explicit (rigorous) definition of "probabilistic classifier"?
>
> A probabilistic classifier is a graph-level classifier which outputs a probability distribution over classes. Formally, this is a function which takes as input a graph and outputs a tuple of probabilities which sum to $1$, where the $i$th element of the tuple gives the probability for the $i$th class.
>
> > In the discussion of almost sure convergence (3.1), what is $\bar\mu$?
>
> Thank you for spotting this typo. It should be $(\mu_n)_{n \in \mathbb N}$.
>
> > From the definition of ``term language'' (Definition 4.1), it seems that it is a high-level summary of components in GNN architectures. I do not see what the essential properties are that lead to key results such as Theorem 5.1. Can authors elaborate on this?
>
> One way to understand this is to relate the proof structure to the definition of the term language. We focus on the denser cases to illustrate this. Lemma 5.4 is proved by induction on the term structure, showing that if a term is composed of simpler terms which converge, then it also converges. For this we treat each case of Definition 4.1 separately, which provides insight into how each contributes to convergence.
>
> For example, the Lipschitz function case uses the smoothness of $F$ to show that if its arguments converge, then its application to those arguments must also converge. The heart of the weighted mean case is a concentration bound showing that a weighted mean is with high probability close to its expected value.
>
> Another way to understand which properties are essential for convergence is to consider extensions of the term language for which it does not hold. For example, if we include a max aggregation operator, we no longer converge a.a.s. to a constant (see the reply to reviewer ETj1).
>
> > For Theorem 5.1, is the almost sure limit a single (deterministic) vector?
>
> It is indeed. We will add a clarificatory sentence to make this explicit.
>
> > I interpret the main findings of the paper as: large graphs (from one of the discussed models) are not distinguishable. Is this correct? Is there any way that the GNN architecture can be modified to overcome such a limitation?
>
> We agree with your interpretation: a GNN will output essentially the same value on all large graphs with high probability.
>
> The question of how to overcome this limitation is intriguing but unfortunately beyond the scope of our current work. One can look towards aggregation functions not included in the term language (such as summation) for potential examples of such non-convergence phenomena. We hope that our work inspires and paves the way for such investigations.
>
> > Node features are assumed to be i.i.d. (3.1). In applications, this is usually not the case. How will non i.i.d. features affect the results?
>
> We agree that this would be an interesting study. There are several challenges from a theoretical perspective. The first is that 'non-i.i.d.' is underspecified: there are many ways in which variables can be dependent. Second, the independence assumption is used in the application of concentration inequalities throughout the proofs, so these would need careful consideration.
>
> We believe that many important cases of dependence (e.g. analogous to the SBM model of dependence for graph structure) can be handled by reducing to our framework. But we leave the verification of this for future work.

---

> > ### Comment · Reviewer_xvBN · 2024-08-08
> >
> > I would like to thank the authors for the detailed rebuttal. Most of my concerns are sufficiently addressed. There are issues beyond the scope of the paper, which are important in my opinion. I suggest the authors consider discussing them as limitations and potential future directions in the revision. Otherwise, I recommend the paper be accepted. I think my original score is fair, which I will maintain. However, I may raise the score later depending on the responses of the other reviewers.

---

> > > ### Author Response · Authors · 2024-08-09
> > > **Thank you for the positive evaluation**
> > >
> > > We thank the reviewer for going through our rebuttal and recommending acceptance of our work. We will discuss the limitations and potential future directions in the revised version of the paper. We are also grateful that the reviewer closely monitors the discussion period and considers to further raise their score accordingly.

---

### Official Review · Reviewer_JXJp · 2024-07-10

**Soundness:** 3
**Presentation:** 3
**Contribution:** 2
**Rating:** 5
**Confidence:** 3

**Summary:**

The paper presents a theoretical study on the expressive power of some popular GNNs (MPNNs with the mean aggregator,  GAT, and GPS+RW) as multi-class classifiers with random featured graphs. It shows that their output converges a.a.s. to a constant function from various random graph models, including Erdős-Rényi, stochastic block model, and Barabási-Albert model. The study includes empirical studies on synthetic and real graphs, confirming the theoretical findings.

**Strengths:**

- The paper presents a formal language that is designed to describe compositions of operators and functions applied to graphs. This language captures the operations of GNNs in the architectures. This can be further applied to other theoretical work on GNNs related to their expressiveness and generalization capabilities.

- The paper empirically validates the theoretical results of both synthetic and real graph datasets.

**Weaknesses:**

- *Unclear Contribution to Expressiveness Analysis.* The expressiveness power is usually used to describe the GNN’s ability to approximate functions, which is essentially to derive generalization bounds or sample complexity. It is not clear how the type of results given in this paper can enable generalization performance analysis.


- *Difference between [a] and the current paper.* [a] presents the output of MPNNs and GCNs as binary classifiers on the ER graph model that will converge to a constant as the graph size increases. This paper proposed that the output of some popular GNNs as multiclass classifiers on several graph models will converge to a constant function as the graph size increases. The current paper can be seen as an extension of [a]. It would be better to discuss the differences between them. Specifically,
  - What is the difference in the proof techniques between lemma 5.4 in this paper and lemma 4.7 in [a]?
  - If we set the number of classes as 2, then will the convergence results of this paper reduce to the results in [a]?

[a] Adam-Day, S. and Ceylan, I., 2024. Zero-one laws of graph neural networks. Advances in Neural Information Processing Systems, 36.

- The experimental results are supported by std (which is good) but without p-value analysis or confidence interval. The statistical significance is not entirely clear.

- Some minors:

  - The authors assume that the node features are in domain [0, 1]^d, and it would be helpful to explain more about the reason for using this assumption in the proof, e.g., how critical is this assumption for the convergence results?
  - Instead of using randomly initialized weights in the experiments, it would be interesting to use learned weights.
  - Line 130, what is the meaning of “A MEANGNN is one where…”.

**Questions:**

Please see the weakness part.

**Limitations:**

The authors discuss the limitation that their proposed theory results are only applicable to the examined graph models and GNN structures employing a mean aggregator, which is adequate. I do not see any negative social impact.

---

> ### Author Rebuttal · Authors · 2024-08-06
>
> Thank you very much for your review and for pointing out the strength and applicability of our term language. We respond to each point below.
>
> > *Unclear Contribution to Expressiveness Analysis*. The expressiveness power is usually used to describe the GNN’s ability to approximate functions, which is essentially to derive generalization bounds or sample complexity. It is not clear how the type of results given in this paper can enable generalization performance analysis.
>
> The results in our work provide strong upper bounds on the capacity of any architecture included in the term language to generalise from smaller graphs to larger ones. So rather than providing quantitative bounds on generalisation, our work gives impossibility results for what tasks are *in principle* learnable by GNNs.
>
> We will add a remark to the introduction clarifying this.
>
> > *Difference between [a] and the current paper*. [...]
> > - What is the difference in the proof techniques between lemma 5.4 in this paper and lemma 4.7 in [a]?
> > - If we set the number of classes as 2, then will the convergence results of this paper reduce to the results in [a]?
>
> In comparison with [a], we note that our proof techniques are more general along two axes. Firstly, while [a] relies on dense ER graphs where convergence is much easier to show, our work goes well beyond this and provides techniques which apply to a much wider range of graph distributions. In particular the sparse ER and BA cases require quite different proof strategies, and we see nothing similar to this in [a]. Moreover in the logarithmic growth case we see a more subtle convergence phenomenon where only most of the node features converge; this is quite beyond the scope of Lemma 4.7 in [a].
>
> Secondly, while the proof techniques in [a] rely on a fixed architecture, our results apply much more generally, as made precise in the definition of the term language. For example, this includes GAT, graph transformers, GPS architectures, and many architectural variations including skip connections. This is a non-trivial distinction, since we must deal with terms representing higher-order functions of nodes, which requires careful treatment.
>
> To be clear, if we set the number of classes to 2 as you suggest, and we do not perform probabilistic classification or regression (the focus of our submission) but rather just Boolean classification, then our result would imply a 'zero-one law' similar to that presented in [a]. However, given that this is a special case, and the much greater generality of our assumptions, we consider that our results go much beyond those of [a].
>
> > The experimental results are supported by std (which is good) but without p-value analysis or confidence interval. The statistical significance is not entirely clear.
>
> In `supplementary-page.pdf` we have regenerated graphs in the original paper using confidence intervals instead of standard deviations (Figure 2). The convergence behaviour of these models remains clear.
>
> > The authors assume that the node features are in domain $[0, 1]^d$, and it would be helpful to explain more about the reason for using this assumption in the proof, e.g., how critical is this assumption for the convergence results?
>
> This assumption is made purely for notational convenience, to avoid using additional letters for the bounds. The proof works just as well with arbitrary bounds.
>
> But note that since linear functions are Lipschitz, our result is already completely general and we don't need to allow arbitrary bounds. Indeed, suppose $\tau(\bar x)$ is a term which we apply to a graph distribution $\mathcal D$ with features in $[a, b]^d$. Modify this distribution to $\mathcal D'$ by applying the function $\bar z \mapsto (\bar z - a) / (b - a)$ to the features. This is now a distribution with features in $[0, 1]$. Modify $\tau$ to $\tau'$ by replacing each $\mathrm H(x)$ by $F(\mathrm H(x))$, where $F$ is the function $\bar z \mapsto (b - a) \bar z + a$. Then evaluating $\tau'$ on $\mathcal D'$ is equivalent to evaluating $\tau$ on $\mathcal D$.
>
> > Instead of using randomly initialized weights in the experiments, it would be interesting to use learned weights.
>
> Thank you for this suggestion. In `supplementary-page.pdf` we include an experiment in which a GCN is trained on the ENZYMES dataset and then analysed asymptotically in the same way as other experiments (Figure 1). As you can see, we observe very similar convergence behaviour to the randomly initialised case.
>
> > Line 130, what is the meaning of “A MEANGNN is one where…”.
>
> We will change this to "A MEANGNN is an MPNN where the aggregate function is the mean."

---

> > ### Comment · Reviewer_JXJp · 2024-08-09
> >
> > I appreciate the author's response.
> >
> > I believe that this paper is above the acceptance bar.

---

> > > ### Author Response · Authors · 2024-08-11
> > >
> > > Thank you for going through our rebuttal and for the positive evaluation.

---

### Official Review · Reviewer_ETj1 · 2024-07-11

**Soundness:** 3
**Presentation:** 2
**Contribution:** 3
**Rating:** 5
**Confidence:** 3

**Summary:**

In this paper, the authors show that GNNs for graph classification applied to several classes of random graphs (with any node features) converges asymptotically almost surely to a constant. This is similar to several recent results in the literature on graphon-like graphs, but with a potentially stronger notion of asymptotic convergence, and with a seemingly very different proofs based on language terms and associated notions.

**Strengths:**

- a very original approach to GNNs on random graphs compared to the literature
- the handling of barabasi-albert graphs, which are very different from the results in the literature which treated of graphons, random geometric graphs, and so on.

**Weaknesses:**

- the presentation of language, terms, and all associated notions is very cryptic for the reader unfamiliar with this specific field (which would be most readers in Neurips). It is very hard to parse the notations and vocabulary, and to relate it to more familiar notions (vectorss, functions, etc.). Some notations seem (?) a bit unconsistent; $x$ sometimes refer to nodes, sometimes to "free variable" (?) and nodes become $u$, and so on. Other notions ("graph-type", "edge negation", "feature controller"...) are very hard to digest

- it is not clear how the authors handle the notion of "class", the figure 1 seems to suggest that there are several classes and that converging to a constant is a limitation, but surely if all the graphs are drawn from the same distribution they should be all in the same class? Or are the labels related to something else? Other works (eg by Levie) precisely look at the ability of GNNs to distinguish between graphs drawn from different distributions

- this work is not the only one to do so, but it mixes two worlds that do not collide in practice: that of graph classification and of asymptotically large graphs. In practice, graph classification tasks are done on small-ish graphs, while large graphs are almost always for node classification.

**Questions:**

- the fact that GNNs converge to a limit object on random graphs has been proved several times before (possibly for less general architectures and distribution of graphs), even with strong non-asymptotic results, in which way an asymptotic convergence is "stronger" as claimed by the authors?

- the authors mention at the end the treatment of max aggregation which may be different; it is indeed treated specifically in their reference [9], could similar technique be employed?

---

> ### Author Rebuttal · Authors · 2024-08-06
>
> Thank you very much for your review and for finding our approach original and acknowledging the handling of Barabasi-Albert graphs in our work — an important difference to existing related works. We respond to each of your comments below.
>
> > the presentation of language, terms, and all associated notions is very cryptic for the reader unfamiliar with this specific field (which would be most readers in Neurips). [...] Other notions ("graph-type", "edge negation", "feature controller"...) are very hard to digest
>
> Thank you for bringing this up. We understand that some parts of the paper are quite technical in nature, however this technicality is unavoidable given the nature of the theoretical work undertaken. We wish to point out that we took pains to ensure that a reader without the relevant background could still understand the key take-aways. For instance, in Section 4.1 we provide many examples of architectures which are included in the term language, and Corollary 5.2 spells out explicitly how the main result applies to GNNs used in practice.
>
> In addition, following the suggestion of reviewer xvBN, we have produced a graphical example of evaluating a term on a small graph (Figure 3 in `supplementary-page.pdf`), to provide some additional intuition for the term language. Following your comments, we will also better explain how the example architectures in Section 4.1 can be captured using terms in our language.
>
> > Some notations seem (?) a bit unconsistent; $x$ sometimes refer to nodes, sometimes to "free variable" (?) and nodes become $u$, and so on.
>
> Thank you for this point; we will add a remark to explain our conventions more explicitly. In the body we reserve $x$ for free variables (which you can think of as similar to letters in algebraic expressions like $y = 3x + 5$) and $u$ for concrete nodes.
>
> > it is not clear how the authors handle the notion of "class", the figure 1 seems to suggest that there are several classes and that converging to a constant is a limitation, but surely if all the graphs are drawn from the same distribution they should be all in the same class? Or are the labels related to something else?
>
> In this work we adopt the common setup used in machine learning, where we have a distribution over labelled inputs, and the goal is to identify the label from the input.
>
> By “different distributions” we assume that the reviewer is referring to the conditional distributions of input graphs conditioned on each label. Such distributions may well be different from each other in a real-world classification task. However, this fact does not interact with our results, since we show convergence for the full distribution on input graphs. In other words, our results show that asymptotically a GNN will not be able to distinguish graphs *no matter what* the conditional distributions are.
>
> > this work is not the only one to do so, but it mixes two worlds that do not collide in practice: that of graph classification and of asymptotically large graphs. In practice, graph classification tasks are done on small-ish graphs, while large graphs are almost always for node classification.
>
> We thank the reviewer for pointing this out. Indeed our results have strong implications beyond graph classification. Lemma 5.4 shows that any node-level GNN converges a.a.s. to a function *only of the input node's features*. In other words "node-level classifiers asymptotically ignore the graph structure". We will highlight this in the paper and discuss the implications which, in our opinion, improves the presentation of our paper and the framing of our results.
>
> While we agree that most current benchmarks contain smaller graphs, we expect that this will likely not always be the case going forward. We therefore consider understanding the asymptotic behaviour of GNNs to be an important theoretical contribution, in that it provides strong bounds on their expressive capacity. Furthermore, we point out that in our experimental evaluation we observed the effect of convergence already for quite small graphs. Such sizes are well within the range of many real-world datasets.
>
> > the fact that GNNs converge to a limit object on random graphs has been proved several times before (possibly for less general architectures and distribution of graphs), even with strong non-asymptotic results, in which way an asymptotic convergence is "stronger" as claimed by the authors?
>
> For example, [9] provides non-asymptotic results, a corollary of which is that under certain conditions the output of a GNN converges *in probability*. Our results show convergence *asymptotically almost surely*, a much stronger notion. Below we provide an example illustrating this.
>
> We are happy to explain the strength of our notion of convergence with respect to any other references the reviewer may have in mind.
>
> > the authors mention at the end the treatment of max aggregation which may be different; it is indeed treated specifically in their reference [9], could similar technique be employed?
>
> As mentioned above, [9] shows a convergence in probability for max aggregation. However, showing asymptotically almost sure convergence is not possible in this case.
>
> For example, using max aggregation one can express the function $F$ that returns $1$ if a graph has a $4$-clique and $0$ otherwise. In some of our typical ER root growth cases, there is a non-trivial proportion of graphs where $F$ returns $0$ and a non-trivial proportion where function $F$ returns $1$ — thus not asymptotically constant. E.g. for edge probability $1/n^{2/3}$, Lynch 1998 showed that asymptotically the frequency of graphs that will have a $4$-clique is around $1-e^{-1/24}$. Shelah and Spencer’s seminal 1988 result shows that for $\mathrm{ER}(n, n^{-\alpha})$ with $\alpha$ rational, first-order logic does not converge in distribution. This implies that there are examples with max aggregation where one does not obtain even convergence in distribution.

---

> > ### Comment · Reviewer_ETj1 · 2024-08-12
> >
> > I thank the authors for taking the time to address my concerns.
> >
> > I still feel that the construction is much too dense and technical for a conference paper, and would greatly benefit from being more pedagogically explained in a longer journal paper. I keep my score as is.

---

### Official Review · Reviewer_6Ly5 · 2024-07-13

**Soundness:** 3
**Presentation:** 3
**Contribution:** 3
**Rating:** 7
**Confidence:** 3

**Summary:**

This paper introduces a term language to express many common GNN architectures as probabilistic classifiers. Using this term language, the authors provide asymptotically almost sure convergence results for dense and sparse random graph models (Erdos-Renyi variants, Barabasi-Albert, Stochastic Block Model). They perform experiments for these graph models as well as one real-world dataset with several different GNN architectures expressible in their term language. Their empirical results align with the presented theory and they can indeed observe convergence to a class distribution as the size of the graphs increases.

**Strengths:**

This contribution seems novel, useful and of interest to the broader graph learning community. Overall, the paper is well-written and, despite its technical nature, relatively easy to follow. The introduction of the term language, which allows for proving asymptotic almost sure convergence in a systematic manner, is very elegant, and the results apply to a broad number of random graph models (dense and sparse) and several state-of the art GNNs.

**Weaknesses:**

The term language is not explained in sufficient detail. E.g., are there any specific assumptions that $\pi$ and $\tau$ need to satisfy? Could you clarify what is expressible within your term language and what is not?

Please find some additional minor remarks below.

* l. 14: ML -> please introduce abbreviation first
* l. 64: what does "import" mean here? (should it be "impact"?)
* l. 127, 132: neighbor vs. neighbour
* l. 137: What is $\mathbf{W}_V$ (in comparison to just $\mathbf{W}$ in GAT)?
* l. 143: dot in sentence (after [12])
* l. 163: What is $k$? The number of nodes?
* l. 165: Consider re-writing this sentence as it is difficult to understand
* Def. 4.3: Missing $G$ subscript?
* l. 233: Sentence seems to be missing something: "Then for every AGG[...] term converges a.a.s. with respect to $(\mu_n)$"
* l. 236: regardless of "the" input graph
* l. 294: "this phenomenon" or "these phenomena"

**Questions:**

1. What happens if we use different (but Lipschitz continuous or smooth) activation functions? E.g., the sine function (while perhaps a peculiar choice) is smooth, but not eventually constant. Is it expressible in your term language?
2. On a related note, could you state more explicitly or provide some intuition on what assumptions are needed for, e.g., activation functions to be expressible in your term language (beyond the discussion, where it is mentioned that we need mean aggregation and activation functions need to be smooth)?
3. Could you say more about the rate of convergence?

**Limitations:**

As briefly mentioned in the **Weaknesses**, the limitations of what can be expressed in the term language are not explicitly stated. While the authors state that the term language is applicable to a wide set of GNN architectures, this is only shown for a subset of them (and for, e.g., GCN, the term language needed to be extended accordingly).

---

> ### Author Rebuttal · Authors · 2024-08-06
>
> Thank you very much for your review and for finding our work novel, useful, easy to follow, and elegant. We respond to each of your comments below.
>
> > The term language is not explained in sufficient detail. E.g., are there any specific assumptions that $\pi$ and $\tau$ need to satisfy? Could you clarify what is expressible within your term language and what is not?
>
> We clarify that we make no additional assumptions on the term language beyond those explicitly stated. We allow *any* Lipschitz function and *arbitrary* combinations of the term language components. In other words, the term language is completely specified by Definition 4.1. This is one of the strengths of our approach, in that it means our results are robust to many architectural choices.
>
> > What happens if we use different (but Lipschitz continuous or smooth) activation functions? E.g., the sine function (while perhaps a peculiar choice) is smooth, but not eventually constant. Is it expressible in your term language?
>
> > On a related note, could you state more explicitly or provide some intuition on what assumptions are needed for, e.g., activation functions to be expressible in your term language (beyond the discussion, where it is mentioned that we need mean aggregation and activation functions need to be smooth)?
>
> All Lipschitz functions are included in the term language, and we do not make any additional assumptions. So in particular the sine function is part of the language (and moreover any Lipschitz continuous activation function you could come up with). Furthermore, any weighted mean whose weights can be computed using terms is part of the language.
>
> > Could you say more about the rate of convergence?
>
> Because our term language is quite rich (e.g. including arbitrary Lipschitz functions), giving explicit bounds on the rate of convergence would involve chasing through many applications of concentration results and other estimations. We expect that this would be quite a complex expression, in terms of parameters like the Lipschitz bounds. We thus confined our analysis of the convergence rates to the empirical observation that GNNs tend to converge quite quickly.
>
> We agree that analysis of convergence rates (as well as work to improve bounds on this) is important, and we leave this to future research.
>
> ---
>
> We thank you for your additional minor remarks, and will make corresponding changes. We comment where appropriate below.
>
> > l. 64: what does "import" mean here? (should it be "impact"?)
>
> We meant 'import' in the sense of "importance or significance". However we recognise that this is somewhat obscure usage, and will change it.
>
> > l. 137: What is $W_V$ (in comparison to just $W$ in GAT)?
>
> Thank you for spotting this typo. We will replace it with $W$.
>
> > l. 163: What is $k$? The number of nodes?
>
> $k$ is an arbitrary (but fixed) integer, which allows for Lipschitz functions which take multiple arguments (e.g. binary summation).
>
> > Def. 4.3: Missing $G$ subscript?
>
> Yes, thank you.

---

> > ### Comment · Reviewer_6Ly5 · 2024-08-10
> >
> > I thank the authors for their response and keep my current score.

---

> > > ### Author Response · Authors · 2024-08-11
> > >
> > > Thank you for going through our rebuttal and for keeping the positive recommendation.

---

### Author Rebuttal · Authors · 2024-08-07

We would like to thank all the reviewers for their time and constructive comments. We are grateful that the reviewers appreciated the novelty of our use of a term language to capture a wide class of GNN architectures, and the fact that our results apply to distributions beyond those typically considered, including  the Barabasi-Albert distribution.

We summarise our new experiments in the global response PDF (`supplementary-page.pdf`) attached to this post. We also briefly summarise our response to a shared concern raised by the reviewers.

## New experiments and figures

1. Following **reviewer JXJp**’s suggestion, we empirically investigated the asymptotic behaviour of a GNN with trained rather than randomly initialised weights (Figure 1). For this we trained a GCN on the ENZYMES dataset. We observe very similar convergence behaviour to the randomly initialised case.
2. Prompted by **reviewer JXJp**’s comment, we regenerated a number of plots with confidence intervals rather than standard deviations (Figure 2).
3. On the suggestion of **reviewer xvBN**, we produced a graphical example of numerically computing a term on a small graph (Figure 3). This provides intuition for how an inductively built term is evaluated on a graph.

## Term language

We emphasise that term language contains all Lipschitz functions and arbitrary combinations of the components in Definition 4.1. This flexibility means that our results apply to a wide class of architectures, even including potential future innovations.

Following **reviewer ETj1**’s comments, we will explain in more detail how the example architectures in Section 4.1 are captured by our term language, to aid the reader’s comprehension of this language concept. We will also include the graphical example of evaluating a term on a small graph (Figure 3) which provides further intuition for this.

We hope that this helps address the Reviewers' concerns, and we look forward to a collaborative discussion period.

---

> ### Comment · Reviewer_6Ly5 · 2024-08-09
>
> Thank you for providing the graphical example in Fig. 3. Could you briefly explain why the degree 1 node (with initial feature $H(x)=-3$) results in $6$, and not $4$?

---

> > ### Author Response · Authors · 2024-08-09
> >
> > Sorry, this is indeed a typo. To clarify, both 6-nodes in the right-most graph in Fig 3 should be 4-nodes instead.  Thank you for pointing this out. We will revise this accordingly.

---

### Decision · Program_Chairs · 2024-09-25

**Decision:**

Accept (poster)

**Comment:**

This paper investigates the expressive power of GNN by analyzing how the predictions of a GNN probabilistic classifier evolve when applied to increasingly larger graphs drawn from various random graph models. The authors demonstrate that the output of these GNNs converges to a constant function, which limits what these classifiers can express uniformly across different graph models. This convergence property is shown to apply to a wide range of GNN architectures and is validated both theoretically and empirically.

The reviewers were mostly positive about the paper, and I recommend accepting it to NeurIPS 2025. Nevertheless, I recommend following the recommendation of the reviewers for the camera-ready version, especially :
- improve the clarity of the paper (see for instance the comment of reviewer ETj1)
- improve the flow of the appendix (of more than 20 pages). The authors did their best to pack as most content as possible in 9 pages, but I agree with the referees that some parts are dense for a conference paper.
- include more details on the experiments as started in the rebuttal.